# EMBO
*reports*

# SGTA associates with nascent membrane protein precursors

Pawel Leznicki* iD & Stephen High** iD

## Abstract

The endoplasmic reticulum (ER) is a major site for membrane protein synthesis in eukaryotes. The majority of integral membrane proteins are delivered to the ER membrane via the co-translational, signal recognition particle (SRP)-dependent route. However, tail-anchored proteins employ an alternative, post-translational route(s) that relies on distinct factors such as a cytosolic protein quality control component, SGTA. We now show that SGTA is selectively recruited to ribosomes synthesising a diverse range of membrane proteins, suggesting that its biosynthetic client base also includes precursors on the co-translational ER delivery pathway. Strikingly, SGTA is recruited to nascent membrane proteins before their transmembrane domain emerges from the ribosome. Hence, SGTA is ideally placed to capture these aggregation prone regions shortly after their synthesis. For nascent membrane proteins on the co-translational pathway, SGTA complements the role of SRP by reducing the co-translational ubiquitination of clients with multiple hydrophobic signal sequences. On this basis, we propose that SGTA acts to mask specific transmembrane domains located in complex membrane proteins until they can engage the ER translocon and become membrane inserted.

**Keywords** co-translational; hydrophobicity; protein quality control; tail-anchor; ubiquitination
**Subject Categories** Membranes & Trafficking; Translation & Protein Quality

## Introduction

As a consequence of the vectorial manner of protein synthesis, the folding and intracellular targeting of nascent polypeptides are often co-translational processes that are facilitated by a number of factors which associate with ribosome–nascent chain complexes (RNCs). Hence, processing of the N-terminal methionine and N-terminal acetylation are mediated by ribosome-bound methionine aminopeptidase and N-acetyl transferases, respectively, co-translational protein folding by various molecular chaperones and co-

translational protein delivery to the endoplasmic reticulum (ER) by the signal recognition particle (SRP) [1]. The ribosome is also a major site for protein quality control, with an estimated ~ 1% of yeast [2] and up to 15% of mammalian [3] nascent polypeptides being ubiquitinated on actively translating ribosomes and potentially degraded by the proteasome. Protein quality control at the ribosome is also triggered in response to perturbations in translation such as ribosome stalling on defective mRNAs. In this case, the ribosome is first split into individual subunits and the 60S subunit-associated, tRNA-bound nascent chain is ubiquitinated by an E3 ligase, listerin, extracted from the 60S subunit by the p97 complex and degraded at the proteasome [4].

The co-translational targeting of proteins destined for the compartments of the secretory pathway is typically mediated by SRP, which recognises hydrophobic ER targeting signals, be they cleavable N-terminal signal sequences or transmembrane domains that can act as signal-anchor sequences, as soon as they emerge from the ribosomal exit tunnel [5]. SRP induces a transient translational pausing and delivers the RNC to the ER membrane where it binds the membrane-tethered SRP receptor (SR), enabling transfer of the RNC to the Sec61 complex which mediates protein translocation into and across the ER membrane [5]. Detailed studies have shown that SRP can be recruited to the translating ribosome before a newly synthesised hydrophobic signal has emerged from the ribosomal exit tunnel, presumably via nascent chain-induced structural rearrangements within the ribosome [6]. Such early recruitment would enhance the binding of SRP to authentic clients in a cellular environment where ribosomes are likely in an excess over SRP [6,7], and/or where levels of SR may be rate limiting [8].

Alternative, post-translational routes for targeting to the ER are used by the group of membrane proteins collectively known as tail-anchored (TA). In the case of TA proteins, their hydrophobic targeting signal, which is located at the extreme C-terminus of the polypeptide, remains hidden within the ribosomal exit tunnel when translation terminates, thereby precluding effective co-translational recognition by SRP. For many TA proteins, their ER targeting signals can be recognised post-translationally by components of the mammalian transmembrane domain recognition complex (TRC), or the equivalent yeast components of the guided entry of tail-anchored proteins (GET) pathway [9,10]. In yeast, TA proteins are first recognised by Sgt2 and then transferred to the ER targeting factor Get3 in a reaction facilitated by Get4/5 [11,12]. A potential

---

School of Biological Sciences, Faculty of Biology, Medicine and Health, University of Manchester, Manchester, UK
*Corresponding author. Tel: ++44 161 275 1526; E-mail: pawel.leznicki@manchester.ac.uk
**Corresponding author. Tel: ++44 161 275 5070; E-mail: stephen.high@manchester.ac.uk

role for molecular chaperones of the Hsp70 family acting upstream of Sgt2 has been suggested by some but not all studies [11,12]. Once bound to its client, Get3 interacts with the Get1/2 transmembrane receptor at the ER membrane allowing for the subsequent membrane insertion of TA proteins. A comparable, albeit more complex, pathway operates in higher eukaryotes with homologues of Sgt2 (SGTA), Get4 (TRC35), Get5 (UBL4A), Get3 (TRC40) and Get1/2 (WRB/CAML) identified [10]. However, higher eukaryotes also have an additional component, Bag6, which together with TRC35 and UBL4A forms the so-called BAG6 complex that facilitates substrate transfer from SGTA to TRC40 [13]. Bag6 also contributes to protein quality control, and it has been implicated in the ubiquitination and proteasomal degradation of defective ribosomal products (DRiPs) [14] and mislocalised membrane and secretory proteins (MLPs) [15,16]. Interestingly, SGTA can both antagonise the Bag6-mediated ubiquitination of MLPs and induce MLP deubiquitination [17,18]. This dual role of Bag6 in both TA-protein targeting to the ER and protein quality control suggests that a mechanism which discriminates between hydrophobic clients destined for ER delivery or selective protein degradation is in operation. This triaging event seems to be determined by the relative affinities of targeting/quality control factors for TA-protein clients, which in turn specify how long such a substrate interacts with a given protein subcomplex [19].

The importance of protein delivery to the ER is underscored by the fact that about a third of eukaryotic proteins are initially targeted there [20–22]. Hence, it is not surprising that multiple ER targeting/insertion pathways exist and that there is significant redundancy between these various routes. Such redundancy is exemplified by the identification of the SND (SRP independent) pathway for protein delivery to the ER in yeast [23], with a mammalian equivalent also suggested [24]. The yeast SND pathway operates co-translationally and is used preferentially by precursors with ER targeting signals of the signal-anchor type located centrally within the polypeptide chain. Strikingly, the SND pathway can compensate for the inactivation of the SRP- or GET-mediated routes in yeast [23] suggesting there is functional interplay between various ER targeting modes. Likewise, in mammals, the model TA-protein Sec61β can exploit the SRP, TRC and SND pathways in parallel [25]. Similarly, it has recently been shown that a subset of TA proteins with relatively hydrophilic signal-anchor sequences can be inserted into the ER membrane via the ER membrane protein complex (EMC) rather than WRB/CAML [26]. These latter TA proteins do not form a stable complex with TRC40 and are proposed to be kept in a membrane insertion competent state through binding to calmodulin and/or SGTA. Although TA-protein targeting and membrane insertion are post-translational events, the fidelity of TA-protein biogenesis would presumably be enhanced by positioning the relevant targeting components at the ribosome to enable early substrate recognition. Hence, yeast Get4/5 [27] and the mammalian BAG6 complex [28] have both been shown to associate with the ribosome, although cross-linking studies suggest that the BAG6 complex does not interact directly with a ribosome-tethered substrate [28]. How such ribosomal recruitment of Get4/5 and the BAG6 complex correlates with an apparently earlier/upstream function for Sgt2/SGTA [11,12,19,29] is unclear.

The role of SGTA as an upstream loading factor in TA-protein delivery [19], its potential function in the EMC-dependent membrane insertion pathway [26], and its interactions with molecular chaperones [30,31] all suggest that SGTA may participate in determining the fate of newly synthesised membrane proteins. Here, we have taken an *in vitro* approach in order to investigate the determinants and order of events that underlie SGTA-substrate interactions. We find that, in addition to TA proteins, SGTA binds a range of membrane protein precursors that contain two or more hydrophobic signal sequences, including single-spanning and multi-spanning membrane proteins. SGTA binding is selective, co-translational and most likely reflects a direct interaction with the nascent polypeptide chain which occurs as soon as a suitable hydrophobic signal emerges from the ribosome. Strikingly, SGTA is recruited to the ribosome before its hydrophobic client emerges from the exit tunnel, suggesting that a priming event coordinates its timely availability. At a functional level, we find that SGTA binding can selectively reduce the co-translational ubiquitination of complex nascent membrane protein precursors, whilst the productive delivery of these RNCs to the ER membrane facilitates SGTA release. Taken together, our data suggest that SGTA can complement the ER targeting role of SRP by shielding specific transmembrane domains (TMDs) in order to enhance the overall fidelity of membrane protein biogenesis.

## Results

### Experimental system

*In vitro* protein synthesis using rabbit reticulocyte lysate (RRL) offers a well-controlled system to study protein biogenesis from a single defined mRNA, and by incorporating $^{35}$S-labelled methionine into the resulting translation products, they are readily detected using phosphorimaging. Furthermore, by using truncated mRNAs of different lengths that each lack a stop codon, it is possible to generate artificial nascent chain intermediates stalled on the ribosome at a defined point of synthesis [32,33]. When combined with site-specific cross-linking, this approach has been successfully used to characterise the interacting partners of a range of nascent polypeptides (e.g. [34–36]). In this study, we have used such a well-established *in vitro* translation system to investigate the interactions of SGTA with newly made membrane protein precursors synthesised in the absence of their target organelle, the ER.

Since SGTA is well characterised as a factor that mediates TA-protein biogenesis, we began by monitoring the interaction of recombinant human SGTA with newly synthesised TA proteins that were translated using RRL. Endogenous levels of mammalian SGTA are proposed to be ~ 1 μM [19], whilst the equivalent yeast component, Sgt2, is ~ 0.5 μM [11,37]. We therefore chose 2 μM recombinant SGTA (HisTrx-tagged) as a physiologically relevant concentration to add to our *in vitro* system prior to protein synthesis, using the HisTrx polypeptide tag alone as a control. We then synthesised selected TA proteins in the presence of HisTrx or HisTrx-SGTA and recovered the two recombinant proteins via immobilised metal affinity chromatography (IMAC) of their respective His-tags. When the resulting pull-downs were analysed, radiolabelled TA proteins were apparent in the imidazole-eluted fraction from translation reactions supplemented with HisTrx-SGTA but not HisTrx alone (Fig 1A, cf. lanes 1–8, see open circles). TA-protein

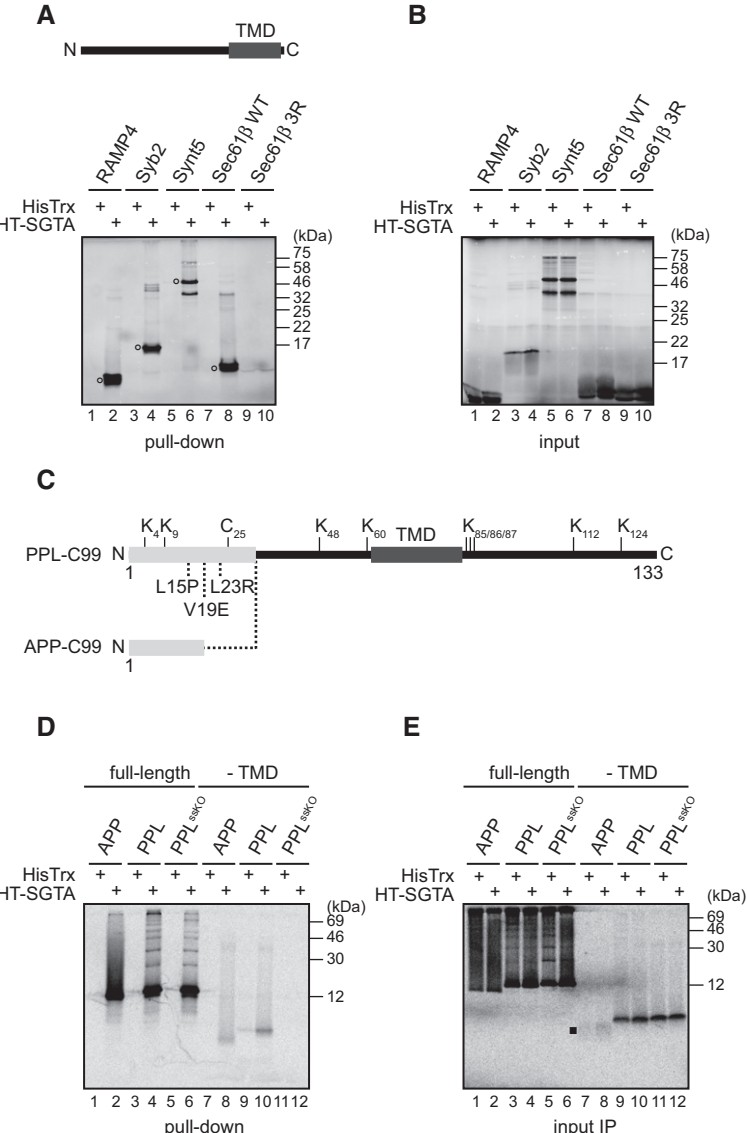

**Figure 1. SGTA binds hydrophobic polypeptides.**

A   Indicated tail-anchored (TA) proteins were *in vitro* translated in rabbit reticulocyte lysate (RRL) using RNAs containing a stop codon. Translations were carried out in the presence of 2 μM HisTrx or HisTrx-SGTA (HT-SGTA), followed by incubation with HisPur Cobalt resin as described in Materials and Methods. Bound proteins were eluted at high imidazole concentration, samples resolved by SDS–PAGE, and results visualised by phosphorimaging. Sec61β 3R carries three Arg residues within Sec61β transmembrane domain (TMD) which abolish its hydrophobic character and was used as a negative control for SGTA binding. A schematic representation of a TA protein is also shown. Open circles indicate TA proteins selectively bound by HisTrx-SGTA. RAMP4—stress-associated endoplasmic reticulum protein 1; Syb2—synaptobrevin 2; Synt5—syntaxin 5.

B   As for (A) but total translation products (input) were resolved by SDS–PAGE and results visualised by phosphorimaging. When total products are analysed, high amounts of haemoglobin in the RRL distort the migration of Sec61β and its variant.

C   A schematic representation of model polypeptides derived from amyloid precursor protein (APP) that were used in the study. The C-terminal 99 amino acids of APP (C99) were fused in frame with the signal sequence derived either from preprolactin (PPL) or APP. Lysine and cysteine residues and amino acids within the preprolactin signal sequence mutated in the PPL^ssKO-C99 construct are indicated whilst the TMD and signal sequences are shown in dark and light grey, respectively.

D   As for (A) but RNAs coding for C99 variants and containing a stop codon introduced either after the complete coding sequence or before the predicted TMD were used.

E   As for (D) but total translation products were subjected to immunoprecipitation with either anti-C99 antibody (for the full-length variants) or with anti-β-amyloid antibody (for products lacking the TMD). A filled square indicates inefficiently translated APP-C99^-TMD immunoprecipitated with anti-β-amyloid antibody.

Source data are available online for this figure.

synthesis, as judged by the amount of total radiolabelled protein product, was directly comparable in the presence of HisTrx and HisTrx-SGTA ruling out any effect of SGTA on protein synthesis *per se* (Fig 1B). However, when a variant of Sec61β bearing three arginine residues within its TMD [28] was synthesised, none of the newly synthesised protein was selectively recovered with

HisTrx-SGTA (Fig 1A, lanes 9 and 10). The introduction of these charged amino acids disrupts the hydrophobic nature of Sec61β TMD [28], and hence taken together, our results indicate that recombinant HisTrx-SGTA binds to newly synthesised, full-length TA proteins via their hydrophobic TMDs. We conclude that *in vitro* protein translation in the presence of recombinant HisTrx-SGTA coupled with IMAC allows us to monitor the authentic association of SGTA with newly made polypeptides.

## SGTA binds membrane proteins with an N-terminal signal sequence

We next asked whether SGTA recognises hydrophobic precursors that can employ the predominant, SRP-dependent co-translational pathway for ER delivery. We have previously shown that the *in vitro* ubiquitination of the C-terminal 99 amino acid residues (C99) of the amyloid precursor protein (APP) fused at its N-terminus to the APP signal sequence is significantly reduced when it is synthesised in the presence of recombinant SGTA, indicative of an SGTA/client interaction [17]. We therefore employed APP-C99 (cf. Fig 1C) as a model precursor to extend our study of SGTA-substrate interactions. A derivative of APP-C99 with its endogenous N-terminal signal sequence substituted for the signal sequence of bovine preprolactin (PPL-C99) [17] was also used, together with a newly constructed PPL-C99 variant bearing three amino acid substitutions within the signal sequence that prevent its recognition by SRP (PPL^ssKO-C99) [36] (Fig 1C). When these substrates were translated in the presence of HisTrx or HisTrx-SGTA, we could clearly detect the selective association of the radiolabelled polypeptides with recombinant SGTA (Fig 1D, lanes 1–6). The introduction of a stop codon before the predicted transmembrane domain ("-TMD"; cf. Fig 1C) generated shorter, faster migrating species, and in the case of APP-C99 and PPL-C99, these truncated polypeptides were also recognised by SGTA (Fig 1D, lanes 7–10), although less effectively than their full-length equivalents (cf. Fig 1D, lanes 1–4). In contrast, when PPL^ssKO-C99 was terminated before its predicted TMD the resulting product was not recovered with SGTA (Fig 1D, lanes 11–12), despite a level of synthesis comparable to that of the equivalent PPL-C99 truncation (Fig 1E, lanes 9–12). On the basis of these results, we conclude that for SGTA to bind a polypeptide that has been released from the ribosome, this substrate must contain an exposed region of hydrophobic character such as a functional ER signal sequence or TMD as previously suggested [12,38,39]. Furthermore, our data suggest that in terms of client hierarchy SGTA may favour hydrophobic TMDs over N-terminal signal sequences (Fig 1D, cf. lanes 1–4 and 7–10).

## SGTA recognises ribosome-bound membrane proteins

We next investigated at which stage during membrane protein biogenesis SGTA can first recognise its substrates. To this end, the same APP-based model precursors were translated *in vitro* using mRNAs that now lacked a stop codon, and the resulting polypeptides were then either stabilised at the ribosome using cycloheximide (CHX) or released with the aminoacyl-tRNA analogue, puromycin [40]. The binding of these two classes of substrates to recombinant SGTA was then investigated as before. In agreement with our previous results (Fig 1), nascent chains that had been

released from the ribosome via a puromycin-induced reaction efficiently co-purified with SGTA but not the HisTrx control (Fig 2A, lanes 7–12). Strikingly, slower migrating radiolabelled species were also recovered with SGTA from CHX-treated samples (Fig 2A, lanes 1–6, filled dots), and the levels of these products were selectively reduced after puromycin treatment, which also increased the levels of unmodified polypeptides (Fig 2A, cf. lanes 1–12). Furthermore, if input reactions were first treated with RNaseA before analysis using SDS–PAGE, these slower migrating species became undetectable (Fig 2B, lanes 1–6) confirming that they are peptidyl-tRNA species. It should be noted that a substantial amount of faster migrating tRNA-free polypeptide species was detected with each precursor even after the RNCs were "stabilised" using CHX treatment (Fig 2A, cf. lanes 1–12). These species most likely reflect the unavoidable hydrolysis of the somewhat labile peptidyl-tRNA bond during the course of our experiments, as detailed in previous studies [41,42]. Since the precise origin of the tRNA-free polypeptide species observed upon treatment with CHX is therefore uncertain, we focused our attention on the slower migrating forms that correspond to tRNA-bound polypeptides. These SGTA-associated species were sensitive to RNaseA digestion (Fig 2C, lanes 1–6) and were selectively recovered using CTAB (Fig 2D), a non-ionic detergent that specifically precipitates peptidyl-tRNA species [28,43]. Taken together, these results show that SGTA is capable of recognising tRNA-bound nascent membrane proteins strongly indicating that SGTA can bind its substrates during their synthesis at the ribosome.

To better understand precisely when a nascent membrane protein is recognised by SGTA, we translated PPL^ssKO-C99 variants using templates lacking a stop codon and terminating either before, or a defined distance after, the predicted TMD (Fig 3A), and then followed SGTA binding to the resulting polypeptides. In this case, we chose PPL^ssKO-C99 because we had already established that the lack of a functional N-terminal signal sequence makes its TMD the only hydrophobic determinant that contributes to SGTA binding in the context of the ribosome-released protein (cf. Fig 1D). As observed before for the released form (Fig 1D), ribosome-stalled PPL^ssKO-C99^-TMD was not recognised by SGTA (Fig 3B, lane 1). Hence, SGTA has no intrinsic ability to recognise nascent polypeptide chains simply as a consequence of their association with the ribosome. Strikingly, SGTA was recruited by nascent ribosome-bound PPL^ssKO-C99 immediately after the synthesis of the TMD region (TMD + 0; see Fig 3B, lane 2), and SGTA continued to recognise these truncated polypeptides as their length increased (Figs 3B, lanes 3–7 and EV1A). We therefore conclude that SGTA can bind to PPL^ssKO-C99 chains immediately after the TMD region has been synthesised.

Given that the length of the PPL^ssKO-C99 TMD region is ~ 25 amino acids, yet the ribosomal exit tunnel accommodates ~ 40 residues [44], SGTA appears to be recruited to the ribosome whilst its substrate TMD is still inside the ribosomal exit tunnel. If that is the case, then we reasoned that SGTA might also be recruited by TA proteins that had been artificially stalled at the ribosome since their C-terminally located TMDs would also be hidden in the ribosomal exit tunnel (cf. Fig 3C). Indeed, when we *in vitro* translated radiolabelled TA proteins using mRNAs lacking a stop codon in the presence of HisTrx or HisTrx-SGTA and carried out IMAC-based pull-downs as before, we could specifically recover tRNA-bound species from HisTrx-SGTA-containing reactions, albeit weakly for

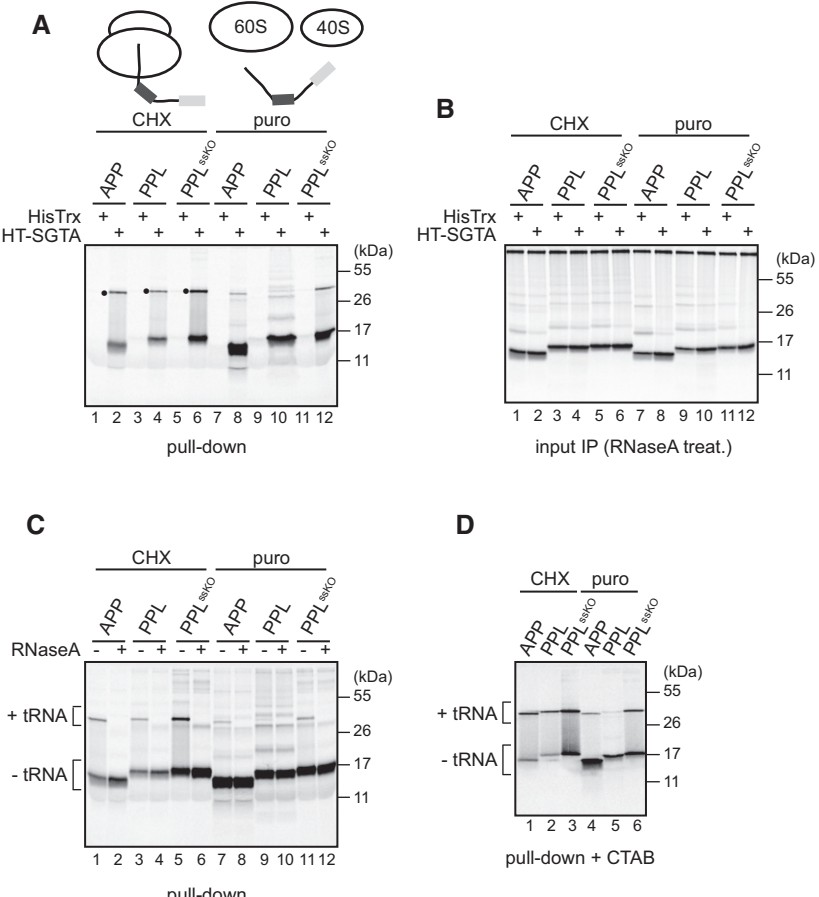

**Figure 2. SGTA binds ribosome-associated nascent membrane proteins.**

A    Full-length polypeptides were translated in the presence of 2 μM HisTrx or HisTrx-SGTA as described in Materials and Methods using RNAs lacking a stop codon and either stabilised at the ribosome with cycloheximide (CHX) or released with puromycin (puro). Reactions were incubated with HisPur Cobalt resin, bound proteins eluted at high imidazole concentration, resolved by SDS–PAGE and visualised by phosphorimaging as described in Materials and Methods. Filled circles indicate slower migrating species of radiolabelled products selectively enriched in CHX-treated samples.

B    As in (A) but total translation reactions were treated with RNaseA and immunoprecipitated with anti-C99 antibody as described in Materials and Methods.

C, D  Following *in vitro* translation of the indicated proteins in the presence of 2 μM HisTrx-SGTA and recovery on HisPur Cobalt resin as described for panel (A), the eluted fractions were subjected to RNaseA treatment (C) or precipitation with hexadecyltrimethylammonium bromide (CTAB) (D) according to Materials and Methods. "−tRNA" and "+tRNA" indicate tRNA-lacking and tRNA-bound protein species, respectively.

Source data are available online for this figure.

Synt5 (Fig 3C, filled dots). As before (Fig 3B), SGTA recruitment to nascent TA proteins also required an intact hydrophobic signal as evidenced by the lack of SGTA binding to ribosome-stalled Sec61β³ᴿ (Fig 3C, lanes 7–10) despite its normal translation (Fig EV1B). Hence, we conclude that the TMD of TA proteins can also recruit SGTA to the ribosome whilst it is located within the ribosomal exit tunnel.

An alternative interpretation of our pull-down results is that SGTA might recognise hydrophobic TMDs that became cytosolically exposed when tRNA-bound nascent chains either fall off the ribosome or become accessible following partial ribosome disassembly. The first possibility seems highly unlikely since tRNA-bound polypeptides translated from mRNAs lacking a stop codon and co-eluted with HisTrx-SGTA from an IMAC resin can subsequently be pelleted through a sucrose cushion, indicative of them being part of the large RNC complex (Fig EV1C, "pelleting" panel).

In the second scenario of partial ribosome disassembly, quality control factors [4] could generate peptidyl-tRNA species bound to the 60S ribosomal subunit, from which a trapped TMD might back slide into the cytosol (cf. Fig 3D, schematic). In order to address this issue, we repeated our pull-down experiments using RRL that had been immunodepleted of the ribosome splitting factor, Hbs1L (Fig EV1D). RRL depleted of Hbs1L is inefficient at mediating ribosome splitting and, as a consequence, shows greatly reduced listerin-dependent nascent chain ubiquitination [45]. We reasoned that if SGTA preferentially recognises 60S bound nascent chains that result from partial ribosome disassembly, then depletion of Hbs1L should reduce SGTA binding (cf. Fig 3D). In practice, we see the opposite effect, and Hbs1L depletion actually enhances the proportion of a stalled tRNA-bound TA protein. Thus, when FLAG-tagged Sec61β is synthesised in Hbs1L-depleted RRL, we observe a statistically significant increase in the level of tRNA-bound Sec61β

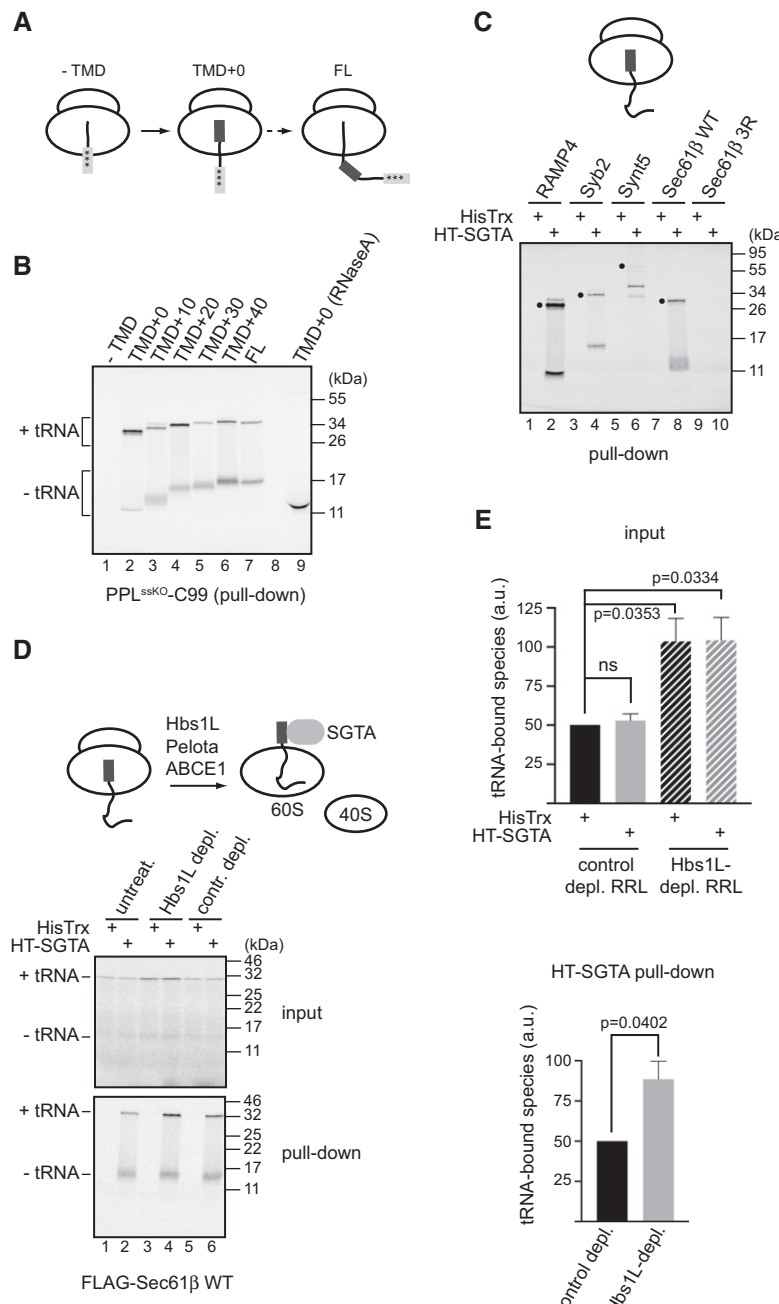

**Figure 3. SGTA is recruited by a transmembrane domain located inside the ribosomal exit tunnel.**

A   A schematic representation of PPL$^{ssKO}$-C99 translation intermediates used in an experiment shown in panel (B). Signal sequence and the TMD are shown in light and dark grey, respectively, whilst three stars indicate mutations of the PPL signal sequence that render it non-functional.

B   Variants of PPL$^{ssKO}$-C99 were translated in the presence of 2 μM HisTrx-SGTA using templates lacking a stop codon and terminating either before ("−TMD") or at indicated amino acid distance after the TMD. Pull-down with HisPur Cobalt resin was carried out as described in Materials and Methods, eluted material resolved by SDS–PAGE, and results visualised by phosphorimaging. tRNA-bound species were verified by treating PPL$^{ssKO}$-C99 variant terminating immediately after the TMD (TMD + 0) with RNaseA (lane 9).

C   As for (B) but translation reactions were carried out using RNAs coding for the indicated TA proteins and lacking a stop codon. Sec61β 3R was used as a control protein without a functional TMD. Filled circles indicate tRNA-bound TA-protein species recovered with HisTrx-SGTA.

D   Rabbit reticulocyte lysate was left untreated ("untreat."), incubated with control immobilised antibodies ("contr. depl.") or immunodepleted of Hbs1L, a factor that together with Pelota and ABCE1 [4], mediates splitting of ribosomes stalled on truncated mRNAs (see diagram). These lysates were used to translate FLAG-tagged Sec61β from an mRNA lacking a stop codon in the presence of 2 μM HisTrx or HisTrx-SGTA, and samples were processed as described for (B).

E   tRNA-bound species of FLAG-tagged Sec61β from experiment shown in panel (D) were quantified and expressed relative to the value obtained for the control-depleted RRL. Shown is the mean with standard error of mean for n = 4 biological replicates. Statistical significance was calculated using unpaired t-test with Welch's correction. ns—not significant.

Source data are available online for this figure.

(Fig 3D, "input" panel, cf. lanes 3 and 4 with 5 and 6, and 3E, "input" panel), consistent with Hbs1L acting as ribosome splitting factor that aids the release of stalled nascent chains. Hence, our data suggest that once any such 60S species are generated, their peptidyl-tRNA bond may be prone to enhanced hydrolysis. Most tellingly, Hbs1L depletion increased the amount of stalled tRNA-bound Sec61β that was recovered with SGTA via pull-down (Fig 3D, "pull-down" panel, cf. lanes 2, 4 and 6, and Fig 3E, "HT-SGTA pull-down" panel). Combined, these data strongly support our hypothesis that SGTA is preferentially recruited to intact 80S ribosomes bearing hydrophobic TMDs that are buried within the exit tunnel.

To determine whether SGTA recruitment to such RNCs may be facilitated by known SGTA interacting partners, we *in vitro* translated PPL-C99$^{FL}$ and Sec61β in the presence of two well-defined SGTA mutants using mRNAs either lacking (Fig EV1E) or containing (Fig EV1F) a stop codon and carried out IMAC-based pull-downs. The SGTA mutants selected were the SGTA$^{D27R/E30R}$ and SGTA$^{K160E/R164E}$ variants, which fail to correctly interact with the BAG6 complex or molecular chaperones and the proteasome, respectively [31,46,47]. We did not detect any obvious changes in substrate recovery with these SGTA variants using either ribosome-stalled or ribosome-released nascent chains (Fig EV1E and F), suggesting that the known binding partners of SGTA are not essential for its recruitment to ribosome-bound nascent chains or its capacity to bind to nascent polypeptide substrates after their release from the ribosome.

To further explore the recruitment of SGTA to RNCs containing ribosome-shielded hydrophobic signals, we next isolated RNCs directly in order to examine their associated factors. To this end, we *in vitro* translated N-terminally FLAG-tagged Sec61β$^{WT}$ and Sec61β$^{3R}$ from mRNAs lacking a stop codon using RRL depleted of Hbs1L, isolated the resulting radiolabelled products by immunoaffinity purification followed by sedimentation of 3xFLAG peptide-eluted material through a sucrose cushion, and investigated the association of endogenous SGTA with these purified RNCs. We observed the specific recruitment of endogenous SGTA to RNCs containing stalled Sec61β$^{WT}$ (Fig 4, lanes 4 and 7) but found that this association was absent when we analysed RNCs generated with the Sec61β$^{3R}$ nascent chain (Fig 4, lanes 5 and 8), despite the comparable recovery of the two FLAG-tagged peptidyl-tRNA species. Importantly, when immunoisolated Sec61β$^{WT}$ and Sec61β$^{3R}$ nascent chains were subsequently pelleted both samples contained comparable levels of 40S and 60S ribosomal subunits (Fig 4, cf. lanes 7 and 8). This finding supports our previous data (Figs 3 and EV1C) which indicated that SGTA is recruited to the intact 80S ribosome and that this recruitment relies on substrate hydrophobicity even when the hydrophobic signal is buried within the ribosomal exit tunnel.

## SGTA interacts with the ribosome-bound nascent chains directly

To investigate whether, once recruited to the ribosome, SGTA may interact with the nascent polypeptide chain, we used a photo-cross-linking approach by incorporating ε-TDBA-Lys (4-(3-trifluoro-methyldiazirino) benzoic acid modified lysine) into the stalled RNCs during their *in vitro* synthesis [35,36,48]. These translation reactions were carried out in the presence of HisTrx-SGTA or HisTrx, samples irradiated with UV light to initiate cross-linking and RNCs isolated

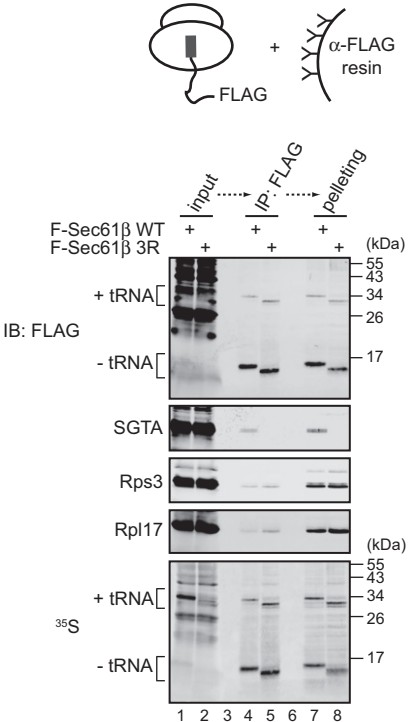

**Figure 4. SGTA recruitment requires an intact transmembrane domain within the ribosomal exit tunnel.**

Sec61β wild-type (WT) and the 3R variant tagged at their N-terminus with the FLAG epitope were translated in RRL from RNAs lacking a stop codon. Ribosome–nascent chain complexes (RNCs) were recovered using anti-FLAG affinity resin, beads were washed, bound proteins eluted with 3xFLAG peptide, and RNCs isolated by pelleting through a sucrose cushion as described in Materials and Methods. Samples were resolved by SDS–PAGE and immunoblotted for endogenous rabbit SGTA, FLAG-tagged translation products, proteins of the 40S (Rps3) and 60S (Rpl17) ribosomal subunits or analysed by phosphorimaging ($^{35}$S). The 3xFLAG peptide-eluted material (lanes 4 and 5) corresponds to ~ 10% of the sedimented RNCs (lanes 7 and 8).

Source data are available online for this figure.

to enrich for adducts formed between the ribosome-bound nascent chain and potential interacting partners. Isolated RNCs were then used in immunoprecipitation reactions in order to identify candidate factors that were close enough to the ribosome-bound nascent chains to be covalently cross-linked to them. PPL-C99 was chosen as a model substrate for these cross-linking studies since its N-terminal signal sequence contains two lysine residues, Lys$^4$ and Lys$^9$ (Fig 1C) that were previously shown to mediate efficient cross-linking of nascent PPL to SRP54 [35].

Irradiation of samples containing stalled PPL-C99$^{-TMD}$ or PPL-C99$^{FL}$ generated high molecular weight species that co-purified with the RNCs and were formed only in the presence of both the ε-TDBA-Lys and UV irradiation (Appendix Fig S1A, see "cross-links"). Upon irradiation, TDBA forms a highly reactive radical that is readily quenched with water, and hence, the formation of a photo-cross-linked adduct with a protein partner is strongly indicative of an extremely close physical proximity/protein–protein interaction [49]. The most prominent PPL-C99$^{-TMD}$ cross-linking product represents an adduct to the SRP54 subunit as confirmed by its immunoprecipitation following denaturation with SDS

(Fig 5A, lanes 3–6) and consistent with the fact that SRP is known to bind signal sequences as they emerge from the ribosome [5]. Unlike SRP54, no cross-linking of PPL-C99$^{-TMD}$ to exogenous SGTA could be detected (Fig 5A, lanes 3–8). More adducts were apparent when stalled PPL-C99$^{FL}$ was analysed (Appendix Fig S1A, cf. lanes 3, 6, 9 and 12) including adducts with SRP54 (Fig 5B, lanes 1–4). However, in this case the addition of HisTrx-SGTA resulted in the formation of a unique HisTrx-SGTA-PPL-C99$^{FL}$ cross-linked product (Fig 5B, lane 8).

In contrast to the clear ~ 60 kDa adduct formed with exogenous human HisTrx-SGTA, we could detect no evidence of cross-linking

products formed between stalled PPL-C99$^{FL}$ and endogenous rabbit SGTA (see Fig 5B, lane 7). We speculated that the antibody used might not recognise rabbit SGTA, and we therefore repeated our cross-linking analysis in the absence of any exogenous human SGTA and used an alternative, chicken anti-SGTA antibody to immunoprecipitate the resulting adducts. Under these conditions, we could now detect a faint cross-linking product that was formed between the stalled, radiolabelled PPL-C99$^{FL}$ and the presumptive endogenous rabbit SGTA orthologue (Appendix Fig S1B). To validate these results, we confirmed the specificity of the chicken anti-SGTA antibody, which we also used in Fig 4, by carrying out Western blotting

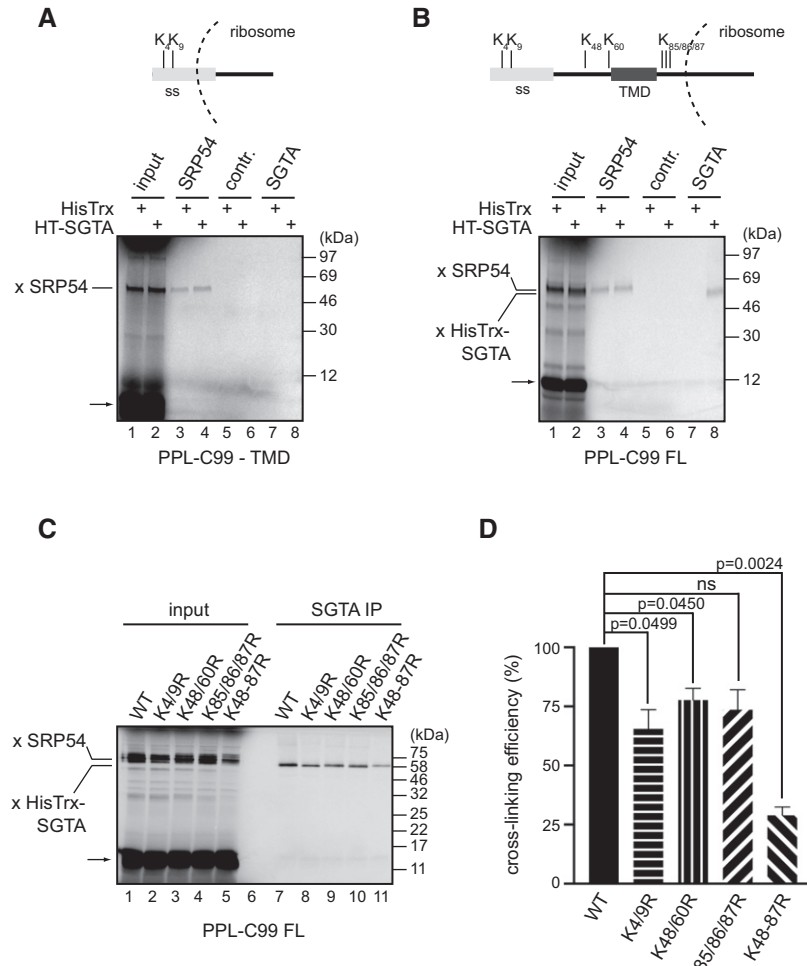

**Figure 5. SGTA binds ribosome-associated nascent chains directly.**

A, B  PPL-C99$^{-TMD}$ (A) and PPL-C99$^{FL}$ (B) were *in vitro* translated in RRL from mRNAs lacking a stop codon in the presence of 2 μM HisTrx or HisTrx-SGTA and ε-TDBA-Lys-tRNA analogue. Reactions were irradiated with the UV light to induce photo-cross-linking, RNCs isolated and adducts immunoprecipitated with mouse anti-SRP54, mouse anti-SGTA or a control antibody as described in Materials and Methods. Samples were resolved by SDS–PAGE and results visualised by phosphorimaging. C-terminal 40 amino acid residues located in the ribosomal exit tunnel are indicated. "x SRP54" and "x HisTrx-SGTA" indicate cross-linking adducts between the translated nascent chain and endogenous SRP54 or recombinant HisTrx-SGTA, respectively. Arrows indicate unmodified translation products.

C    PPL-C99$^{FL}$ variants lacking a stop codon and carrying the indicated Lys to Arg substitutions were *in vitro* translated in the presence of 2 μM HisTrx-SGTA. Reactions were processed as for panels (A and B) using chicken anti-SGTA antibody for immunoprecipitation of SGTA adducts.

D    Cross-linking adducts between ribosome-stalled PPL-C99$^{FL}$ lysine mutants and recombinant HisTrx-SGTA shown in panel (C) were quantified, any differences in translation efficiency accounted for and cross-linking efficiency expressed relative to the wild-type (WT) protein. Shown is the mean with standard error of mean for $n = 3$ biological replicates. Statistical significance was calculated using unpaired *t*-test with Welch's correction.

Source data are available online for this figure.

on human HeLa cell lysate and RRL with the antibody preincubated with purified, recombinant human HisTrx-SGTA (Appendix Fig S2A).

To understand why human and rabbit SGTAs might be differently recognised by our anti-SGTA antibody, we attempted to compare their amino acid sequences. However, to our surprise, we found that the only annotated rabbit orthologue of SGTA (cf. Ref. [38]) (G1SX57) has just ~ 56% amino acid identity with the human protein (Appendix Fig S2B). To further characterise the endogenous rabbit SGTA-like protein that forms adducts with ribosome-stalled PPL-C99$^{FL}$ (Appendix Fig S1B and Fig EV2) and co-purifies with stalled Sec61$\beta^{WT}$ (Fig 4), we used a mass spectrometry-based approach. In one set of experiments, we identified by in-gel trypsin digestion a protein of ~ 38 kDa that was specifically recovered from RRL using immobilised chicken anti-SGTA antibody (Appendix Fig S2C). As a complementary approach, we also used bacterially expressed, purified and then immobilised Sec61$\beta$ and its variant lacking the TMD, and hence devoid of any hydrophobic determinants, as baits to carry out pull-downs from RRL. We then used in-gel trypsin digestion to identify the ~ 38 kDa protein that was specifically bound to full-length Sec61$\beta$ and eluted in a buffer supplemented with Triton X-100 (Appendix Fig S2D). We had previously used this latter approach to successfully identify Bag6 as a biogenesis factor for TA proteins [38]. Both sets of experiments identified specific peptides that could be unambiguously assigned to human SGTA and that were quite distinct from G1SX57 sequence (hereafter called rabbit SGTB) (Appendix Fig S2B). Hence, our results (this study) together with a previous study that identified SGTA as a binding partner of a mitochondrial membrane protein synthesised in RRL [50], both suggest that a rabbit orthologue of SGTA exists at the protein level but has not to date been annotated in the rabbit genome. Our conclusion is further supported by a multiple sequence alignment that shows the clear conservation of SGTA across various mammalian species, including American pika (Ochotona princeps) which, like rabbit, belongs to the taxonomic order Lagomorpha (Appendix Fig S3A). All of these species, including Ochotona princeps, also have a distinct orthologue of human SGTB which shows high conservation with rabbit SGTB (Appendix Fig S3B). Interestingly, rabbit SGTB (G1SX57) was also identified as a binding partner of immobilised Sec61$\beta$ during our mass spectrometry-based analysis (see Appendix Fig S2B and D) suggesting that some degree of functional redundancy between SGTA and SGTB might exist (see below).

The efficient photo-cross-linking of SRP54 to PPL-C99 (Fig 5A and B, lanes 3 and 4) is consistent with the role of SRP in the early recognition of ER-destined nascent chains via the binding of the SRP54 subunit to their N-terminal signal sequence as described for preprolactin [35]. To better understand which region of ribosome-bound PPL-C99$^{FL}$ is recognised by SGTA, we performed cross-linking experiments using stalled nascent chains with a restricted number of lysine residues. We found that the most significant reduction in the efficiency of exogenous SGTA cross-linking to stalled PPL-C99$^{FL}$ was observed when all of the lysine residues that flank the TMD of PPL-C99$^{FL}$ were replaced with arginines (K48-87R) (Fig 5C, cf. lanes 7–11 and Fig 5D). On this basis, we propose that SGTA selectively associates with ribosome-bound PPL-C99$^{FL}$ via its hydrophobic TMD region (cf. Fig 5B, schematic). Interestingly, the cross-linking of exogenous SGTA to a PPL-C99$^{FL}$ variant that only

contains photoprobes in its N-terminal signal sequence was not completely abolished (Fig 5C, cf. lanes 7 and 11, and 5D). Similarly, mutating lysine residues located within the PPL signal sequence resulted in statistically significant reduction in the efficiency of exogenous SGTA cross-linking to stalled PPL-C99$^{FL}$ (Fig 5C, cf. lanes 7 and 8, and 5D). Hence, it would appear that SGTA can associate with both regions of hydrophobicity present in PPL-C99$^{FL}$ (cf. Fig 5B schematic). We speculate that SGTA may be able to bind N-terminal signal sequences should SRP dissociate, for example as a consequence of chain extension (see also Discussion). We could also recapitulate the cross-linking of PPL-C99$^{FL}$ to SRP54 and/or SGTA using a range of bifunctional chemical cross-linking reagents (Fig EV2A). Importantly, when the reactions were carried out in the absence of exogenous SGTA we could detect a DSS-dependent cross-linking adduct between ribosome-stalled PPL-C99$^{FL}$ and endogenous SGTA (Fig EV2B, lane 7), which mirrors our finding when using a photo-cross-linking reagent (Appendix Fig S1B). When cross-linking of ribosome-stalled PPL-C99$^{FL}$ was carried out in the presence of 2 and 20 μM HisTrx-SGTA, DSS-dependent adducts with SRP54 appeared unaffected whilst SGTA cross-linking was enhanced (Fig EV2B, lanes 4–9). On this basis, we conclude that there is no direct competition between SRP and SGTA for nascent chain binding.

Having established that SGTA can interact with an artificial nascent membrane protein substrate that has two hydrophobic signals, we next investigated whether such an interaction could be observed with naturally occurring membrane protein precursors. To this end, we analysed whether SGTA interacts co-translationally with a range of proteins that contain at least two hydrophobic signals (N-terminal signal sequence and/or TMDs), and which would be expected to exploit the SRP-dependent route for targeting to the ER (Fig EV3A). We chose a range of candidate proteins such that when synthesised as stalled nascent chains, their downstream hydrophobic signal(s) is located either within the ribosome, just outside the exit tunnel or well away from the ribosome. These topological variations allowed us to test what types of membrane protein precursor may be capable of recruiting SGTA to their ribosome-bound nascent chains. Strikingly, when we in vitro translated these polypeptides from mRNAs lacking a stop codon in the presence of HisTrx or HisTrx-SGTA and carried out IMAC-based pull-downs, we could detect the specific binding of each of these nascent chains to SGTA (Fig EV3B–G). In each case, amongst the translation products recognised by SGTA were RNaseA-sensitive peptidyl-tRNA species that represent ribosome-associated nascent chains. On this basis, we conclude that SGTA can be recruited to the ribosome–nascent chain complex by a wide range of substrates that are predicted to rely on co-translational, SRP-dependent targeting to the ER membrane. We used bifunctional cross-linking reagents to further analyse the interaction of SGTA with one of these substrates, CD247, chosen because it contains several lysine and cysteine residues within or close to its N-terminal signal sequence and TMD (Fig EV3A). As for PPL-C99$^{FL}$ (cf. Fig 5), analysis of the purified RNCs showed that both endogenous SRP54 and exogenous SGTA were selectively cross-linked to ribosome-stalled CD247 indicative of their close proximity to the nascent chain (Fig EV3H and I). Based on these data, we conclude that the recruitment of SGTA to nascent PPL-C99$^{FL}$ accurately mirrors the behaviour of naturally occurring membrane protein precursors that contain multiple hydrophobic signals.

## SGTA binding is reversible and reduces co-translational ubiquitination

The functional interaction of nascent precursor proteins with their dedicated delivery factors is normally transient, and such factors are typically released once the appropriate host membrane/organelle is provided *in vitro* [5,51]. To address the nature of SGTA-PPL-C99 binding, we repeated photo-cross-linking reactions with ribosome-stalled PPL-C99[FL] in the presence or absence of canine pancreatic microsomes. The inclusion of ER-derived microsomes resulted in a qualitative reduction in the formation of the nascent chain-SGTA adducts with both exogenous (Fig 6A) and endogenous (Fig 6B) SGTA. Similar results were obtained using the bifunctional reagent DSS, and we used this approach to quantify the ER membrane-dependent reduction in HisTrx-SGTA adduct formation (Fig 6C–E). We found that in the presence of ER-derived membranes the efficiency of DSS-mediated cross-linking between HisTrx-SGTA and ribosome-stalled PPL-C99[FL] was reduced by ~ 60% as compared to reactions lacking ER-derived microsomes (Fig 6E). This suggests that once the hydrophobic region(s) of a nascent polypeptide substrate can be correctly accommodated by the ER membrane, its interaction with SGTA is lost. We therefore conclude that the binding of SGTA to a nascent chain is normally transient, and as such, it may therefore constitute an intermediate step during productive membrane protein biogenesis.

During TA-protein targeting to the ER membrane, SGTA captures the nascent chain as it is released from the ribosome and by interacting with the BAG6 complex enables its TA-protein client to be loaded onto TRC40 [19]. To investigate any potential role of the BAG6 complex in the membrane-dependent release of SGTA-bound nascent chains, we repeated the cross-linking analysis of PPL-C99[FL] associated factors using Bag6-depleted RRL (Fig EV4). Our results indicate that Bag6, the core component of the BAG6 complex, is dispensable for the membrane-stimulated reduction of SGTA cross-linking to ribosome-bound nascent chains (Fig EV4A, cf. lanes 1 and 2 with lanes 3 and 4). On this basis, we conclude that the mechanism by which SGTA interacts with SRP-dependent precursors at the ribosome and its subsequent co-ordinated release are distinct from its established roles in TA-protein biogenesis [8,16].

We have previously reported that SGTA can inhibit ubiquitination and stimulate deubiquitination of MLPs without affecting the global homeostasis of the ubiquitin–proteasome system [17,18]. Since nascent chains can be co-translationally ubiquitinated at the ribosome [2,3], we asked whether the association of SGTA with the hydrophobic region(s) of ribosome-stalled nascent chains might affect their ubiquitination status. Hence, we *in vitro* translated selected membrane protein precursors previously shown to co-translationally associate with SGTA (cf. Fig EV3). For this experiment, we employed mRNAs lacking a stop codon and used RRL that had also been supplemented with HA-tagged ubiquitin and either HisTrx or HisTrx-SGTA. To specifically focus on co-translational events, we then isolated the resulting RNCs and analysed the ubiquitination of the respective ribosome-associated nascent chains by immunoprecipitation (Fig 7A). Quantification of each of the clearly resolved ubiquitinated nascent chain species showed that for F11R and TMEM174 the inclusion of exogenous SGTA significantly reduced the amount of all ubiquitinated precursors that were recovered (Fig 7A, lanes 9–12; Fig 7B). Interestingly, in case of CD247

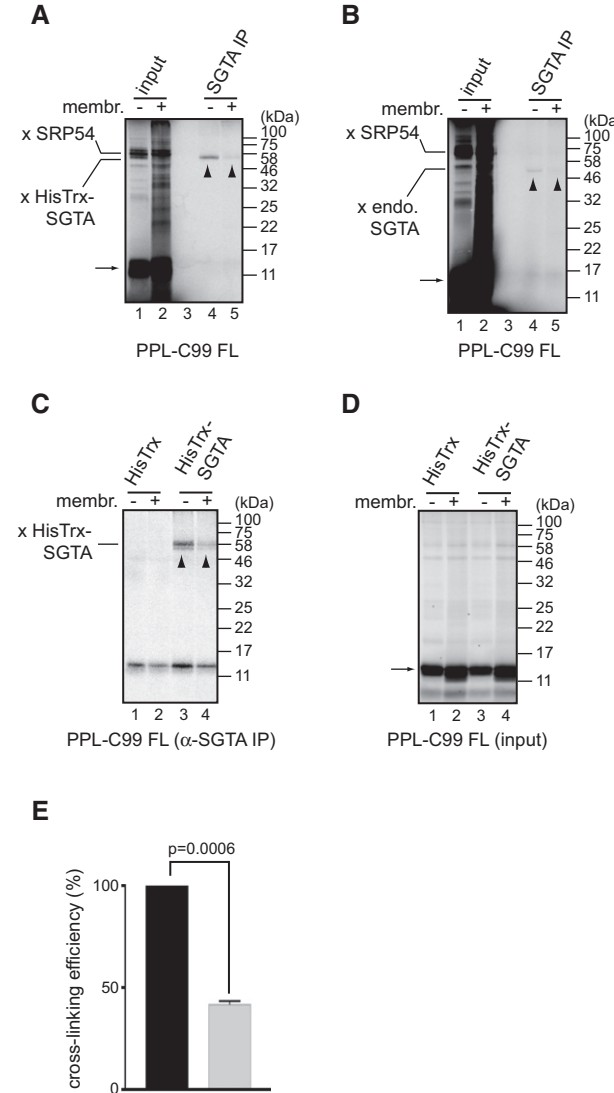

**Figure 6. SGTA binding is reduced in the presence of ER-derived membranes.**

A, B   PPL-C99[FL] lacking a stop codon was *in vitro* translated in RRL in the presence of canine pancreatic microsomes or buffer control in reactions supplemented with ε-TDBA-Lys-tRNA analogue. Reactions were carried out in the presence (A) or absence (B) of 2 μM HisTrx-SGTA. Samples were subjected to UV light-induced photo-cross-linking and processed as described for Fig 5. "x SRP54" and "x HisTrx-SGTA" indicate cross-linking adducts between the translated nascent chain and endogenous SRP54 or recombinant HisTrx-SGTA, respectively, whilst "endo. SGTA" indicates cross-linking adducts with endogenous SGTA. Cross-linking products between stalled PPL-C99[FL] and SGTA are also indicated with arrowheads, whilst arrows indicate unmodified translation products.

C, D   As for panel (A) but chemical cross-linking using DSS reagent was carried out.

E   DSS-mediated HisTrx-SGTA cross-linking efficiency to PPL-C99[FL] in the absence or presence of ER-derived microsomes was quantified. For each repeat, intensity of the cross-linked SGTA adduct in the absence of microsomes was set as 100%. Shown is the mean with standard error of mean for n = 3 biological replicates. Statistical significance was calculated using unpaired *t*-test with Welch's correction.

Source data are available online for this figure.

exogenous SGTA inhibited only the formation of nascent chains bearing three ubiquitin moieties suggesting that nascent chain conformation might affect SGTA action (Fig 7A and B).

To understand the mechanistic basis of SGTA action, we then analysed the co-translational ubiquitination of stalled TMEM174 in the presence of HisTrx or HisTrx-SGTA and an inhibitor of deubiquitinating enzymes (DUBs), ubiquitin aldehyde (Ub-Ald), which we have previously shown to block the SGTA-induced deubiquitination of MLPs [17]. Even in the presence of this DUB inhibitor, exogenous SGTA still significantly reduces the ubiquitination of stalled TMEM174 (Fig EV5A and B). We therefore conclude that SGTA most likely acts by binding the TMD of a ribosome-stalled nascent chain and thereby by shielding it from ubiquitination, rather than by actively promoting nascent chain deubiquitination. If this model is correct, then we speculated that an SGTB could have a similar function, since rabbit SGTB was also identified as a binding partner of immobilised TA protein (cf. Appendix Fig S2). We therefore bacterially expressed and purified both human and rabbit SGTBs and repeated TMEM174 co-translational ubiquitination assay in the presence of these recombinant proteins. We found that both human and rabbit SGTB were as effective in preventing stalled TMEM174 ubiquitination as human SGTA (Fig EV5C and D) supporting our model that SGTA binding is sufficient to protect the nascent chain from co-translational ubiquitination. On the basis of all these findings, we propose that the binding of SGTA to nascent membrane protein precursors can supplement the actions of SRP by preventing co-translational ubiquitination prior to the integration of presynthesised transmembrane domains (see Fig 7C).

## Discussion

Previous studies indicated that SGTA and its yeast orthologue, Sgt2, bind TA proteins post-translationally either directly [12,39] and/or via a reaction facilitated by Hsp70 chaperones [11]. The physiological order of events that enable the binding of mammalian SGTA to its clients has not, however, been addressed in any detail. Using an *in vitro* translation system, we provide evidence that SGTA can be recruited to the ribosome whilst the TMD of a TA-protein client is still located within the ribosomal exit tunnel. This previously unanticipated recruitment of SGTA to the ribosome prior to the exit of a hydrophobic signal from the exit tunnel is reminiscent of SRP

recruitment to a translating ribosome [6]. Furthermore, as for SRP [6], such a ribosomal localisation of SGTA would make it ideally placed to capture newly synthesised TA proteins following translation termination, nascent chain release and the emergence of the hydrophobic TA region into the cytosol. This in turn may favour TA-protein handoff from SGTA to TRC40 via the previously defined pathway that is facilitated by the BAG6 complex [19]. Our model strongly supports the well-documented role of SGTA/Sgt2 as an early acting factor during TA-protein biogenesis [11,12,19,29] and suggests SGTA can begin to act as the TA region enters the ribosomal exit tunnel. Whilst this early ribosomal engagement of SGTA requires the synthesis of a hydrophobic signal, it appears to be independent of the BAG6 complex (cf. Ref. [28]). Hence, neither the removal of Bag6 by immunodepletion nor the use of an SGTA mutant incapable of binding the BAG6 complex impact upon SGTA recruitment. Similarly, binding of SGTA to RNCs is not affected by mutations in its TPR region which mediates interaction with molecular chaperones of the Hsp70 and Hsp90 families [30,31], suggesting that ribosomal recruitment of SGTA is independent of these chaperones. In short, the ribosomal recruitment of SGTA would ensure that it has early access to potential TA-protein clients which may in turn help to increase the fidelity of their post-translational delivery to the ER.

An *in vitro* study using a yeast translation system suggested that the binding of Sgt2 to its TA substrates is strictly post-translational and concluded that Sgt2 is not recruited to RNCs that are synthesising TA proteins [27]. However, it is important to note that yeast lack the Bag6 protein, which is a key component of the mammalian TA-protein targeting cascade, and promotes the selective ubiquitination of MLPs [15,16,28,38]. Hence, in higher eukaryotes TA proteins may run the risk of entering the BAG6 mediated protein quality control/ubiquitination pathway rather than being handed off to TRC40 for ER delivery. In higher eukaryotes, the decision between a biosynthetic and degradative fate for a TA protein appears to be dictated by the relative affinities of SGTA, the BAG6 complex and TRC40 for its clients [19]. In this light, the early binding of SGTA to a TA protein would provide demonstrated protection against premature ubiquitination [17,18].

Unexpectedly, our results show that SGTA can also be recruited by the nascent, ribosome-bound precursors of membrane proteins that are either known or predicted to engage SRP. On the basis of our data, we conclude that, where substrates follow a

**Figure 7. SGTA reduces the co-translational ubiquitination of nascent membrane proteins.**

A  Indicated precursor proteins were translated *in vitro* using RRL supplemented with 20 μM HA-tagged ubiquitin and 2 μM of either HisTrx or HisTrx-SGTA. RNCs were isolated, ubiquitinated nascent chains recovered using anti-HA agarose, and samples resolved by SDS–PAGE and analysed by phosphorimaging. Ubiquitinated precursor protein species were assigned based on SDS–PAGE electrophoretic mobility and are indicated in red. A schematic diagram of the proteins used is shown, and the ~ 40 residues of the nascent chain located in the ribosomal exit tunnel are indicated. "ss"—signal sequence, "TMD"—transmembrane domain.

B  Individual, clearly resolved ubiquitinated species of the indicated proteins (labelled 1–4) were quantified and their intensity in HisTrx-SGTA containing samples shown relative to samples supplemented with HisTrx control protein. Shown is the mean with standard error of mean for *n* = 3 biological replicates. Statistical significance was calculated using unpaired *t*-test with Welch's correction. ns—not significant.

C  A schematic model for SGTA action during membrane protein biogenesis. A signal sequence or the first TMD of a nascent chain is bound by SRP as it leaves the ribosomal exit tunnel. If the elongation arrest is inefficient or the distance between the first and second hydrophobic signals short, SGTA is recruited to the ribosome by a downstream signal located inside the ribosomal exit tunnel (route i). As translation continues, SGTA binds directly to the exposed hydrophobic signal and can reduce co-translational ubiquitination of the nascent chain. Upon arrival to the ER membrane, both SRP and SGTA dissociate from the nascent chain, which is inserted into the ER lipid bilayer. In this model, inefficient recruitment of SGTA (route ii) might result in increased co-translational ubiquitination of nascent chains and possibly their proteasomal degradation.

Source data are available online for this figure.

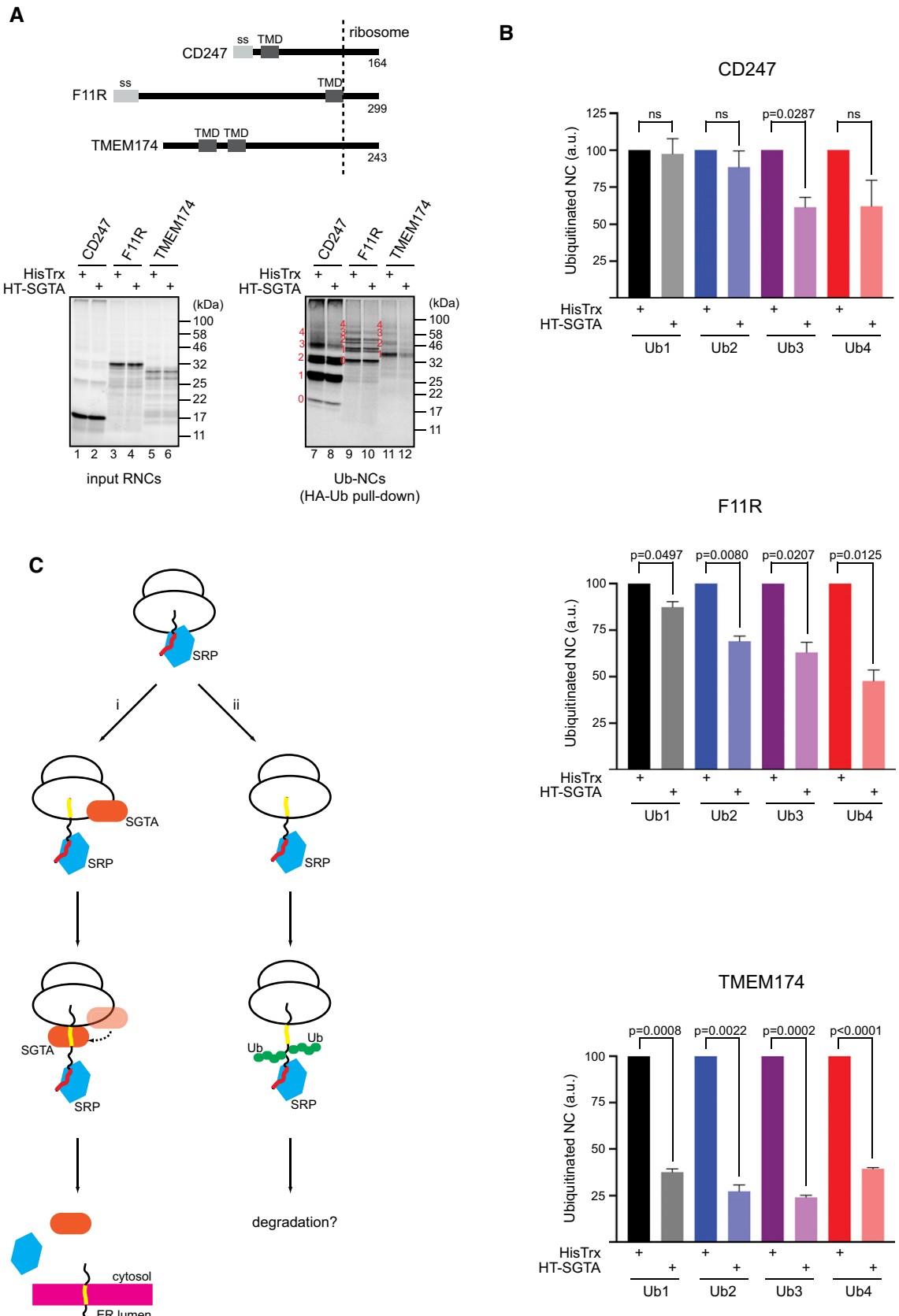

**Figure 7.**

co-translational delivery route, once the most N-terminal ER targeting signal has recruited SRP additional hydrophobic domains that emerge whilst the polypeptide is still being delivered to and/or engaging the ER translocon may be potential clients for SGTA (Fig 7Ci).

The mammalian cytosol contains orders of magnitude more SGTA (~ 1 μM, [19]) than SRP (~ 5–10 nM, [51]). Hence, the binding of SGTA to selected downstream hydrophobic transmembrane domains, such as those closely following an initial targeting sequence or functioning purely as a stop-transfer signal [52], could limit the need to use SRP in a non-targeting, chaperone-like, role. In contrast, when a downstream hydrophobic signal is needed to reinitiate ER protein translocation, for example as a consequence of a long cytoplasmic loop, then an additional round of SRP binding may be required [53], consistent with the ability of SRP to bind internal TMDs in both yeast and bacteria [54,55].

Since both eukaryotic SRP [6] and SGTA (this study) can be recruited to the ribosome via a TMD that is still located inside the ribosomal exit tunnel, we speculate that nascent membrane proteins may induce structural rearrangements within the ribosome that expose binding sites for both components. The simultaneous binding of SRP and SGTA to the same RNC resembles the interplay previously observed between bacterial SRP and trigger factor [56]. Importantly, SGTA does not appear to compete with SRP for binding to the first hydrophobic signal as it emerges from the ribosome, but is only detected after additional chain extension has occurred (this study). Such binding likely reflects an association between SGTA and N-terminal signal sequences that have either been skipped by SRP as observed for several of yeast precursor proteins [54], or from which SRP has dissociated as the nascent chain length increases [57–59]. Furthermore, as for SRP [60], we find that the presence of ER membranes reduces the amount of SGTA associated with nascent membrane protein precursors suggesting that it acts at a point prior to the membrane integration step. Whether SGTA is actively displaced from an RNC by a membrane-localised protein, the potential identity of which is currently unknown will be a subject of future studies with possible candidates including the SRP receptor, the Sec61 complex and the EMC.

SGTA promotes the deubiquitination of MLPs [17,18], and we now show that it can also reduce the ubiquitination of ribosome-stalled nascent membrane protein precursors, in this case seemingly by shielding TMDs that would otherwise be exposed to the cytosol during the co-translational biosynthetic route adopted by many ER targeted substrates (cf. Fig 7i and ii). Such premature TMD exposure may arise when SRP-induced elongation arrest is inefficient [8], through ineffectual engagement of the Sec61 translocon [61], or during the normal biogenesis of particular multispanning membrane proteins (cf. Fig EV3A). This may be particularly relevant for precursors with two hydrophobic signals that are separated by only a short hydrophilic linker, as exemplified by TMEM174 which has only 13 aa between its two predicted TMDs (cf. Fig 7A). A consequence of such an architecture is that a second hydrophobic signal is already synthesised and located within the ribosomal exit tunnel as SRP binds to the first hydrophobic signal (cf. Ref. [62]). However, the separation of two hydrophobic signals alone does not seem to be the sole determinant of potential SGTA substrates. This is illustrated by the fact that ubiquitination of CD247, whose signal sequence is located close to the TMD, is only marginally affected by

SGTA whilst ubiquitination of F11R, whose signal sequence is separated from the TMD by a relatively long linker, is significantly reduced by SGTA. It therefore seems likely that factors such as nascent chain conformation and/or translation rate also play important roles in specifying SGTA clients. Future studies using cell-based assays such as ribosome profiling will help to clarify the nature and extent of such ribosome-associated SGTA clients. Nevertheless, on the basis of our *in vitro* studies we propose that at least some nascent membrane proteins may rely on SGTA to prevent the cytosolic exposure of downstream TMDs and thereby forestall the ubiquitination and/or aggregation of these nascent chains until they engage the ER translocon.

On the basis of the results outlined above, we have formulated the following model for the action of SGTA: a hydrophobic targeting signal such as an N-terminal signal sequence is bound by SRP soon after it emerges from the ribosomal exit tunnel, whilst downstream hydrophobic signals, for example TMDs, can begin to recruit SGTA from within the ribosomal exit tunnel (Fig 7Ci). These additional regions of hydrophobicity are then bound directly by SGTA as they become solvent exposed, and this interaction can limit co-translational nascent chain ubiquitination (cf. Fig 7Cii). Following SRP-mediated delivery to the ER membrane, SGTA may either be actively displaced or cycle on and off its client TMDs (cf. Ref. [19]) prior to membrane insertion. In a wider context, our findings add to the growing body of evidence that the biogenesis of membrane proteins in eukaryotes is complex and that an unexpectedly broad range of cellular components is employed, often in a substrate selective fashion [63–65].

# Materials and Methods

### Materials

APP-C99 in pcDNA5, PPL-C99 in pcDNA3.1 and SGTA in pHisTrx were previously described [17] whilst PPL[ssKO]-C99 and PPL-C99 lysine mutants in pcDNA3.1, SGTA[D27R/E30R] and SGTA[K160E/R164E] in pHisTrx were prepared by site-directed mutagenesis. Plasmids coding for proteins used in Fig EV3 were obtained from SinoBiological. Recombinant proteins were purified as before [17,39,66]. Mouse anti-SGTA (clone 47-B), anti-SRP54 (clone 30) and chicken control IgY antibodies were purchased from SantaCruz Biotechnology whilst mouse anti-β-amyloid (clone BAM-10), mouse control antibodies, anti-FLAG affinity resin, anti-HA agarose and 3xFLAG peptide from Sigma. Rabbit anti-Hbs1L antibody (A305-395A) was from Bethyl Laboratories whilst chicken anti-SGTA, rabbit anti-C99 and rabbit anti-Bag6 antibodies were made to order [17]. Rabbit reticulocyte lysate (RRL) for *in vitro* protein translation was from Promega, chemical cross-linkers from Thermo Fisher Scientific, and ε-TDBA-Lys-tRNA was made essentially as previously described [48].

### *In vitro* transcription, translation and pull-down assays

Templates for *in vitro* transcription were generated by PCR using appropriate primers incorporating either a stop codon or up to two valine residues previously reported to stabilise nascent chains bound to the ribosome [67] with constructs terminating before the predicted TMD containing two additional methionines towards the

end of the sequence to improve radiolabelling. RNA was prepared as previously described [66] and used in an *in vitro* translation reaction using RRL supplemented with 1 mCi/ml [$^{35}$S]methionine, amino acid mix lacking methionine and 2 μM indicated recombinant proteins. Translation of proteins using mRNAs containing a stop codon was carried out for 10 min at 30°C, further translation initiation blocked with 0.1 mM aurintricarboxylic acid, and reactions incubated for another 20 min at 30°C. Ribosome-stalled C99 variants and TA proteins were generated by translating proteins for 5 min at 30°C followed by the addition of aurintricarboxylic acid to 0.1 mM and further incubation for 5 min at 30°C. For proteins used in experiments shown in Figs 7, EV3 and EV5 translation was carried out for 10 min, aurintricarboxylic acid added and reactions incubated for 15 min at 30°C. Ribosome-stalled nascent chains were stabilised with 2.5 mM cycloheximide (CHX) (Sigma) for 5 min on ice or released by the addition of 1 mM puromycin (Sigma) and incubation for 5 min at 37°C.

Samples were diluted 5.5-fold with buffer A (50 mM HEPES-KOH, pH 7.5, 300 mM NaCl, 10 mM imidazole, 10% (v/v) glycerol) supplemented with 5 mM $MgCl_2$ when working with stalled nascent chains, and incubated with pre-equilibrated HisPur Cobalt resin (Thermo Scientific) for 30 min at 4°C for stalled nascent chains or 2 h for stop codon containing ones. Beads were washed extensively with buffer A (supplemented with 5 mM $MgCl_2$ for stalled nascent chains) and bound material eluted with buffer A supplemented with 200 mM imidazole and 5 mM $MgCl_2$ for stalled nascent chains. RNaseA treatment of the eluate was performed by adding RNaseA to a final concentration of 250 μg/ml and incubating for 5 min at 37°C whereas ribosome–nascent chain complexes were isolated by ultracentrifugation (385,000 × g, 8 min, 4°C) through a 0.5 M sucrose cushion in buffer B (25 mM HEPES-KOH, pH 7.5, 80 mM KOAc, 1 mM Mg(OAc)$_2$) [68]. Selective precipitation of tRNA-bound protein species was carried out by adding 10 volumes of 2% hexadecyltrimethylammonium bromide (CTAB) in 10 mM NaOAc, pH 5.0 to the eluate from HisPur Cobalt beads followed by the addition of 10 volumes of 0.5 M NaOAc, pH 5.0 supplemented with 0.2 mg/ml bacterial tRNA (Sigma). After 10-min incubation at 30°C, the precipitate was isolated by centrifugation (16,000 × g, 10 min, RT) and the pellet resuspended in SDS sample buffer.

FLAG-Sec61β WT and 3R variant were *in vitro* translated for a total of 35 min at 30°C using mRNAs lacking a stop codon and RRL depleted of Hbs1L splitting factor (see below). CHX-stabilised reactions were diluted ~ 4 fold in buffer C (20 mM HEPES-KOH, pH 7.5, 150 mM KOAc, 2 mM Mg(OAc)$_2$) supplemented with 2.5 mM CHX, 1 mM PMSF and protease inhibitor cocktail, and incubated with pre-equilibrated anti-FLAG M2 affinity resin (Sigma) for ~ 150 min at 4°C. Beads were washed extensively with buffer C supplemented with 2.5 mM CHX and bound proteins eluted twice with buffer C supplemented with 2.5 mM CHX and 0.5 mg/ml 3xFLAG peptide for 20 min at 4°C. The eluates were combined and RNCs isolated by ultracentrifugation (385,000 × g, 8 min, 4°C) through a 0.5 M sucrose cushion in buffer B. The pelleted RNCs were then resuspended in SDS sample buffer.

All samples were resolved by SDS–PAGE and results visualised by phosphorimaging using Typhoon FLA 7000 (GE Healthcare). Images were processed and band intensity quantified using AIDA software.

## Cross-linking, immunoprecipitation and *in vitro* ubiquitination assay

Protein translation was carried out as described above in RRL supplemented with ε-TDBA-Lys-tRNA [35,36,48], 1 mCi/ml [$^{35}$S] methionine, amino acid mix lacking methionine and lysine and, unless indicated otherwise, 2 μM indicated recombinant proteins. CHX-stabilised reactions were irradiated with UV light for 12 min using Blak-Ray B-100AP High-Intensity UV Lamp, spun down through 0.5 M sucrose cushion in buffer B (385,000 × g, 8 min, 4°C), resuspended in buffer B and treated with 250 μg/ml RNaseA for 5 min at 37°C. An input sample was directly mixed with SDS sample buffer whilst the remaining material denatured with 1% (w/v) SDS for 30 min at 37°C. Reactions were diluted 5-fold with Triton X-100 IP buffer (10 mM Tris-Cl, pH 7.5, 140 mM NaCl, 1 mM EDTA, 1% (v/v) Triton X-100) supplemented with 20 mM cold Cys/Met mix, pansorbin, 1 mM PMSF and complete protease inhibitor cocktail (Sigma) and incubated for 1 h at 4°C. Following a pre-clearing step (16,000 × g, 5 min, 4°C), the supernatant was split, indicated antibodies added and reactions incubated overnight. Antibodies were then recovered on Protein A Sepharose or chicken IgY precipitating resin (GenScript) by 2-h incubation at 4°C, beads washed extensively with Triton X-100 IP buffer, and bound proteins eluted with SDS sample buffer and resolved by SDS–PAGE followed by phosphorimaging analysis. For immunoprecipitation directly from total translation reactions, samples were processed as described above but without the SDS denaturation step.

*In vitro* ubiquitination was carried out by translating indicated precursor proteins using RRL supplemented with 2 μM HisTrx or indicated HisTrx-tagged proteins and 20 μM HA-tagged recombinant human ubiquitin (Boston Biochem). 5 μM ubiquitin aldehyde (Boston Biochem) or control buffer was included where indicated (Fig EV5A and B). Samples were treated with 10 mM NEM for 5 min at 30°C to block further ubiquitination and deubiquitination, and RNCs were stabilised with 2.5 mM CHX for 5 min on ice. Reactions were spun down through 0.5 M sucrose cushion in buffer B (385,000 × g, 8 min, 4°C), resuspended in buffer B and treated with 250 μg/ml RNaseA for 5 min at 37°C. An input sample was directly mixed with SDS sample buffer whilst the remaining material denatured with 1% (w/v) SDS for 30 min at 37°C. Reactions were diluted 5-fold with Triton X-100 IP buffer supplemented with 20 mM cold Cys/Met mix, 1 mM PMSF and complete protease inhibitor cocktail (Sigma), insoluble material spun down (16,000 × g, 5 min, 4°C) and anti-HA agarose (Sigma) added to the soluble fraction. Following overnight incubation at 4°C, beads were extensively washed with Triton X-100 IP buffer, bound proteins eluted with SDS sample buffer, resolved by SDS–PAGE and visualised by phosphorimaging. Species corresponding to ubiquitinated nascent chains were identified in anti-HA immunoprecipitation samples as a ladder of more slowly migrating species, and these were quantified using AIDA software.

Bag6 and Hbs1L immunodepletion from RRL was carried out as previously described [17]. Chemical cross-linking was carried out as described above for photo-cross-linking but proteins were *in vitro* translated in RRL supplemented with 1 mCi/ml [$^{35}$S]methionine, 2 μM indicated recombinant proteins and amino acid mix lacking methionine, and CHX-stabilised chains were cross-linked by adding

freshly prepared cross-linkers to 0.5 mM final concentration and incubating reactions for 30 min at 10°C.

## Statistical analysis

Radiolabelled protein species were quantified using AIDA software, and relative intensities of matched samples calculated in Microsoft Excel. GraphPad Prism was then used to generate graphs and quantify statistical significance using unpaired *t*-test with Welch's correction.

## Identification of the endogenous RRL protein that interacts with nascent chains

In order to identify the endogenous RRL protein that forms cross-linking adducts with stalled PPL-C99$^{FL}$, chicken anti-SGTA antibodies were covalently coupled to chicken IgY precipitating resin (GenScript) and incubated with pre-cleared RRL for ~ 150 min at 4°C followed by extensive washing with RIPA buffer (50 mM Tris-Cl, pH 7.5, 150 mM NaCl, 1% (w/v) NP-40, 1 mM EDTA, 1 mM EGTA, 0.1% (w/v) SDS, 0.5% (w/v) sodium deoxycholate). Bound proteins were eluted with 50 mM glycine, pH 2.5 in two steps for 5 min each at 4°C, eluates combined and eluted proteins precipitated by adding trichloroacetic acid (TCA) to a final concentration of 20% (w/v). Samples were incubated on ice for 1 h, insoluble material spun down (16,000 × *g*, 30 min, 4°C), and pellets washed twice with ice-cold acetone and resuspended in SDS sample buffer.

Alternatively, purified recombinant Sec61β full-length or its variant lacking the TMD was covalently coupled to Ultralink Biosupport (Thermo Fisher Scientific) and used in pull-down reactions from RRL as previously described [38]. Triton X-100-eluted material was TCA precipitated and processed as described above. All samples were resolved by SDS–PAGE and stained with Coomassie Brilliant Blue R-250.

Gel bands were subjected to in-gel tryptic digestion by the University of Bristol Proteomics Facility using a DigestPro automated digestion unit (Intavis Ltd.). The resulting peptides were fractionated using an Ultimate 3000 nano-LC system in line with an LTQ-Orbitrap Velos mass spectrometer (Thermo Scientific). In brief, peptides in 1% (v/v) formic acid were injected onto an Acclaim PepMap C18 nano-trap column (Thermo Scientific). After washing with 0.5% (v/v) acetonitrile, 0.1% (v/v) formic acid peptides were resolved on a 250 mm × 75 μm Acclaim PepMap C18 reverse phase analytical column (Thermo Scientific) over an 80 min organic gradient (1–50% solvent B over 55 min, 50–90% B over 0.5 min, held at 90% B for 5 min and then reduced to 1% B over 0.5 min), with a flow rate of 300 nl/min. Solvent A was 0.1% formic acid, and Solvent B was aqueous 80% acetonitrile in 0.1% formic acid. Peptides were ionised by nano-electrospray ionisation at 2.1 kV using a stainless steel emitter with an internal diameter of 30 μm (Thermo Scientific) and a capillary temperature of 250°C. Tandem mass spectra were acquired using an LTQ-Orbitrap Velos mass spectrometer controlled by Xcalibur 2.1 software (Thermo Scientific) and operated in data-dependent acquisition mode. The Orbitrap was set to analyse the survey scans at 60,000 resolution (at m/z 400) in the mass range m/z 300–2,000, and the top twenty multiply charged ions in each duty cycle selected for MS/MS in the LTQ linear ion trap. Charge state filtering, where unassigned precursor ions were not selected for fragmentation, and dynamic exclusion (repeat count, 1; repeat duration, 20 s; exclusion list size, 500) were used. Fragmentation conditions in the LTQ were as follows: normalised collision energy, 40%; activation *q*, 0.25; activation time 10 ms; and minimum ion selection intensity, 500 counts.

The raw data files were processed and quantified using Proteome Discoverer software v2.1 (Thermo Scientific) and searched against the UniProt Homo sapiens database (downloaded October 2019; 150786 sequences), the UniProt Oryctolagus cuniculus database (downloaded October 2019; 23017 sequences) and a common contaminants database using the SEQUEST algorithm. Peptide precursor mass tolerance was set at 10 ppm, and MS/MS tolerance was set at 0.6 Da. Search criteria included oxidation of methionine (+15.9949) as a variable modification and carbamidomethylation of cysteine (+57.0214) as a fixed modification. Searches were performed with full tryptic digestion, and a maximum of two missed cleavages were allowed. The reverse database search option was enabled, and all data were filtered to satisfy false discovery rate (FDR) of 5%.

**Expanded View** for this article is available online.

## Acknowledgements
We would like to thank Martin Pool (University of Manchester, UK), Blanche Schwappach (University of Goettingen, DE), Lisa Swanton (University of Manchester, UK) and Simon Hubbard (University of Manchester, UK) for their help and feedback during the preparation of this manuscript. We are grateful to members of the University of Bristol Proteomics Facility (University of Bristol, UK) for their assistance with mass spectrometry-based protein identification. This work was supported by a Wellcome Trust Investigator Award in Science (204957/Z/16/Z) to SH.

## Author contributions
PL performed the experiments. PL and SH wrote the manuscript and designed and analysed the experiments.

## Conflict of interest
The authors declare that they have no conflict of interest.

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
