## [Review Process File · EMBO Reports]

SGTA associates with nascent membrane protein precursors

Pawel Leznicki and Stephen High

Review timeline:

Submission date:	10 July 2019
Editorial Decision:	19 August 2019
Revision received:	3 December 2019
Editorial Decision:	24 February 2020
Revision received:	25 February 2020
Accepted:	28 February 2020

Transaction Report:

1st Editorial Decision

19 August 2019

Thank you for the submission of your research manuscript to our journal. We have now received the full set of referee reports that is copied below.

As you will see, the referees acknowledge that the findings are potentially interesting. However, referees 2 and 3 also point out several technical concerns and have a number of suggestions for how the study should be strengthened, and I think that all of them should be addressed. It will be important to exclude that SGTA binds released nascent chains, to verify that tagged human SGTA is functional, that endogenous SGTA interacts with RNCs and to clarify the variability in ubiquitination that was also pointed out by referee 1. Moreover, the limitations of the approach and current dataset should be mentioned and discussed.

Given these constructive comments, we would like to invite you to revise your manuscript with the understanding that the referee concerns (as detailed above and in their reports) must be fully addressed and their suggestions taken on board. Please address all referee concerns in a complete point-by-point response. Acceptance of the manuscript will depend on a positive outcome of a second round of review. It is EMBO reports policy to allow a single round of revision only and acceptance or rejection of the manuscript will therefore depend on the completeness of your responses included in the next, final version of the manuscript.

Revised manuscripts should be submitted within three months of a request for revision; they will otherwise be treated as new submissions. Please contact us if a 3-months time frame is not sufficient for the revisions so that we can discuss the revisions further.

- 2) individual production quality figure files as .eps, .tif, .jpg (one file per figure). Please download our Figure Preparation Guidelines (figure preparation pdf) from our Author Guidelines pages <https://www.embopress.org/page/journal/14693178/authorguide> for more info on how to prepare your figures.
- 3) a .docx formatted letter INCLUDING the reviewers' reports and your detailed point-by-point responses to their comments. As part of the EMBO Press transparent editorial process, the point-by-point response is part of the Review Process File (RPF), which will be published alongside your paper.
- 4) a complete author checklist, which you can download from our author guidelines (<<https://www.embopress.org/page/journal/14693178/authorguide>>). Please insert information in the checklist that is also reflected in the manuscript. The completed author checklist will also be part of the RPF.
- 5) Please note that all corresponding authors are required to supply an ORCID ID for their name upon submission of a revised manuscript (<<https://orcid.org/>>). Please find instructions on how to link your ORCID ID to your account in our manuscript tracking system in our Author guidelines (<<https://www.embopress.org/page/journal/14693178/authorguide#authorshipguidelines>>)
- 6) We replaced Supplementary Information with Expanded View (EV) Figures and Tables that are collapsible/expandable online. A maximum of 5 EV Figures can be typeset. EV Figures should be cited as "Figure EV1, Figure EV2" etc... in the text and their respective legends should be included in the main text after the legends of regular figures.
- For the figures that you do NOT wish to display as Expanded View figures, they should be bundled together with their legends in a single PDF file called *Appendix*, which should start with a short Table of Content. Appendix figures should be referred to in the main text as: "Appendix Figure S1, Appendix Figure S2" etc. See detailed instructions regarding expanded view here: <<https://www.embopress.org/page/journal/14693178/authorguide#expandedview>>
 - Additional Tables/Datasets should be labeled and referred to as Table EV1, Dataset EV1, etc. Legends have to be provided in a separate tab in case of .xls files. Alternatively, the legend can be supplied as a separate text file (README) and zipped together with the Table/Dataset file.
- 7) We would also encourage you to include the source data for figure panels that show essential data. Numerical data should be provided as individual .xls or .csv files (including a tab describing the data). For blots or microscopy, uncropped images should be submitted (using a zip archive if multiple images need to be supplied for one panel). Additional information on source data and instruction on how to label the files are available <<https://www.embopress.org/page/journal/14693178/authorguide#sourcedata>>.
- 8) Our journal encourages inclusion of *data citations in the reference list* to directly cite datasets that were re-used and obtained from public databases. Data citations in the article text are distinct from normal bibliographical citations and should directly link to the database records from which the data can be accessed. In the main text, data citations are formatted as follows: "Data ref: Smith et al, 2001" or "Data ref: NCBI Sequence Read Archive PRJNA342805, 2017". In the Reference list, data citations must be labeled with "[DATASET]". A data reference must provide the database name, accession number/identifiers and a resolvable link to the landing page from which the data can be accessed at the end of the reference. Further instructions are available at <<https://www.embopress.org/page/journal/14693178/authorguide#referencesformat>>.
- 9) Regarding data quantification:
- Please ensure to specify the name of the statistical test used to generate error bars and P values, the number (n) of independent experiments underlying each data point (not replicate measures of one sample), and the test used to calculate p-values in each figure legend. Discussion of statistical methodology can be reported in the materials and methods section, but figure legends should contain a basic description of n, P and the test applied.

IMPORTANT: Please note that error bars and statistical comparisons may only be applied to data obtained from at least three independent biological replicates. If the data rely on a smaller number of replicates, scatter blots showing individual data points are recommended.

- Graphs must include a description of the bars and the error bars (s.d., s.e.m.).
- Please also include scale bars in all microscopy images.

10) As part of the EMBO publication's Transparent Editorial Process, EMBO reports publishes online a Review Process File to accompany accepted manuscripts. This File will be published in conjunction with your paper and will include the referee reports, your point-by-point response and all pertinent correspondence relating to the manuscript.

I look forward to seeing a revised version of your manuscript when it is ready. Please let me know if you have questions or comments regarding the revision.

REFeree REPORTS

Referee #1:

A small but significant percentage of proteins are delivered to the ER via C-terminal tail-anchor (TA) sequences, and in recent years the machinery required for their targeting and insertion into the ER has been elucidated. In this manuscript, Leznicki and High use a series of biochemical techniques to better characterize one member of the targeting complex, SGTA, which functions with TRC40/Get3, a conserved ATPase that plays a central role in this process. Specifically, Leznicki and High show that SGTA associates with TA proteins as well as with an SRP-dependent membrane protein, a modified form of C99 that harbors an exogenous signal sequence. More surprisingly, SGTA appears to identify proteins with membrane spans prior to the time that a transmembrane domain has emerged into the cytosol. Negative data suggest that Bag6 does not contribute to this association. These data reflect discoveries on the action of SRP in targeting nascent proteins to the ER, i.e., SRP also associates with ribosomes and engages polypeptides prior to the emergence of a transmembrane domain. Moreover, the authors report that SGTA can bind to both a signal sequence and a transmembrane domain, albeit with different efficiencies, in the same protein. Next, Leznicki and High found that the inclusion of microsomes in the *in vitro* translation reaction significantly decreased the extent of SGTA association with the signal sequence-containing C99 protein, and this reaction-performed with this SRP-dependent substrate-also appeared to be independent of Bag6. Finally, consistent with its role as a chaperone-like protein, SGTA protects nascent proteins from the protein quality control machinery, namely ubiquitination, which can otherwise target aberrant nascent proteins for degradation.

Overall, this is an intriguing study that is nicely presented. The quality of the data is high, the results were confirmed using a range of crosslinking/pull-down conditions, and alternative interpretations of the data (e.g., ribosome dissociation) were excluded. Substrates containing specific mutations were also used to refine the conclusions. Nevertheless, it will ultimately be important to complement the results obtained in this study with cell-based systems, such as ribosome profiling combined with SGTA precipitations. This attack is beyond the scope of the current study, but perhaps any caveats from a solely *in vitro* study should be spelled out. In a related vein, the number of SRP-dependent substrates was limited, and again in the future the use of a higher throughput strategy would better cement the authors conclusions. Also, the extent of ubiquitination in Fig. 7B varied significantly, and perhaps the authors can speculate on the reason for this variation and make predictions based on which substrates might exhibit a magnified requirement for SGTA as a nascent chain protector. One final open question regards the relationship of other SGTA-associated chaperones and the EMC in

these early SRP and SGTA-dependent events. Perhaps this can be discussed or incorporated into the figure.

Other comments:

1. The authors should include a Table or text to define the substrates used in this paper. What is RELT, F11R, etc?
2. In Fig. EV2, please define the arrowhead and arrow.

Referee #2:

Comments to the Author

EMBOR-2019-48835V1
Leznicki and High,

"SGTA associates with nascent membrane protein precursors."

How newly made transmembrane proteins are accurately assembled into the ER membrane? This point has been receiving the most attention as an important issue for a long time of decades. In this manuscript, Drs. Leznicki and High provided a new insight on this issue from the viewpoint of SGTA, a component of cytosolic protein quality control machinery.

Using an *in vitro* translation system, the authors investigated the interactions of SGTA with newly made membrane protein precursors. They showed that SGTA is capable of recognizing a range of nascent membrane proteins. Amongst the translation products recognized by SGTA were peptidyl-tRNA species that represent ribosome-associated nascent chains. Notably, SGTA recruitment to the ribosome relies on the existence of TMD even when the hydrophobic sequence is still located within the ribosomal exit tunnel, possibly through inducing structural rearrangements of ribosomes. SGTA binds directly to the exposed hydrophobic signal and prevent the cytosolic exposure of TMDs, and thereby prevent the ubiquitination and/or aggregation of these nascent chains until they engage the ER translocon. Indeed, SGTA dissociate from the nascent chain when the client was successfully inserted into the ER lipid bilayer. Such a previously unanticipated recruitment of SGTA to the ribosome would ensure that it has early access to client membrane proteins which may in turn help to increase the fidelity of their post-translational delivery to the ER.

I found this manuscript contains really interesting ideas that how SRP and SGTA collaborate together to support the successful synthesis of multiple membrane spanning proteins. With full of insightful viewpoint, I found the publication of this manuscript at this moment is really timely. Therefore, I recommend this manuscript for publication after adding some experiments to support their main claims. My specific comments were provided as follows.

Specific comments;

(1)

The authors proposed that SGTA recognizes ribosome-associated transmembrane protein precursors to prevent the ubiquitination and/or aggregation of these nascent chains until they engage to the ER. I found this idea very interesting. To further support the proposed function of SGTA, I recommend to immunodeplete the endogenous SGTA protein from RRL with their anti-SGTA antibody. Since RRL contains endogenous SGTA (as shown in Figs 6B and EV2), and authors had performed immunodepletion of Hbs1L (Fig. 3E) and BAG6 (Fig. EV5A) from RLL, these points can be clarified relatively easily. As the idea is so important, I believe that SGTA depletion should be necessary before claiming SGTA is critical for quality control of nascent membrane proteins.

(2)

In Fig 7A, the description in the text does not correspond well with the actual data. The authors noted that the inclusion of exogenous SGTA significantly reduced the amount of ubiquitinated precursor (p14, line 27). However, I can see a very small decrease of ubiquitinated substrates with SGTA addition, both in the case in F11R and TMEM174. Are these small differences of

polyubiquitination of the membrane protein physiologically relevant? Why the co-translational ubiquitination of CD247 was not affected significantly by the addition of HT-SGTA, even though two hydrophobic signals (SS and TMD) of this protein are separated by a short linker as in the case of TMEM174?

(3)

Related to my previous comment, authors quantified the accumulations of ubiquitinated precursor species derived from several membrane proteins in Fig 7A and 7B. I would like to ask you whether this assay includes inhibitors for proteasomes and deubiquitination enzymes in the RLL. Since RLL contains large amounts of these enzymes, the presence or the absence of these inhibitors might affect the interpretation of the results greatly. I recommend to add these inhibitors in the RLL and precipitation buffer for these kind of polyubiquitin pull-down assay.

Other minor points:

(4)

It is helpful if authors indicate the position of 11 kDa marker in the Fig. 1, as like in Figs 3E and 6, since RAMP4 and Sec6a beta migrated around this size.

(5)

What is "x" in xSRP54 in Fig 5 and Fig 6? I assume these mean "cross linked with PPL-C99", since the sum of the molecular weights of SRP54 and PPL-C99 gives that of xSRP54. Please define the exact meaning of this abbreviation in the corresponding figure legends.

(6)

The data in Fig. 5C need quantifications. Can you see statistically significant differences between K4/9R and K48-87R?

(7)

In the reference list, I found a number of style inconsistency (all characters were capitalized in Ref. 40, for an example). Please amend them.

Referee #3:

Leznicki and High report interesting findings, which shed new light on the ER targeting function of SGTA. Most importantly, the authors present data, which show that SGTA i) can interact with ribosome-bound nascent chains (RNCs) of ER-targeted proteins containing C-terminal tail anchors or internal transmembrane domains (TMDs); ii) binds to these RNCs even when the TMD is still inside the ribosomal tunnel; and iii) may interact with RNCs, even if SRP is bound.

These findings are novel and have important implications for the whole ER targeting field.

However, there are two general problems with the experimental set up, which, in my view, need to be clarified to justify the conclusions drawn by the authors.

1. It is not firmly established that SGTA binds to ribosome-bound nascent chains, rather than to released nascent chains. Studies analysing the binding of factors to RNCs standardly sediment RNCs to remove any spontaneously released polypeptides prior to further analysis. Such unwanted release occurs inevitably in every translation system! This sedimentation step is missing in the experiments performed in this study. This is a critical issue that needs to be addressed.

2. There needs to be some evidence that tagged human SGTA (HisTrx-SGTA) is able to functionally replace untagged rabbit SGTA. The authors state that the two proteins are not highly homologous. Possibly HisTrx-SGTA, which is added to translation reactions in excess when compared to the other GET pathway components, does "something" endogenous SGTA, which is bound to partner proteins/chaperones, would never do? In this context, it is essential to improve Fig. 4 (see also below) and Fig. EV2, which should become part of the main part of the manuscript. If I understand correctly the authors suggest that chicken anti-SGTA can IP rabbit SGTA? This should be confirmed via a control blot. Should be possible using the anti-human SGTA AB, which seemingly recognizes rabbit SGTA on Western Blots (Fig. 4). A control blot or appropriate

reference should be added to show that the relevant band in Fig. 4 is indeed rabbit SGTA.

Some more comments and suggestions are detailed below.

Fig. 1

The data convincingly show the well characterized post-translational interaction between SGTA and ER-targeted proteins. With respect to the data shown below, however, I wonder what is the larger species detected in the Synb2, Sec61b (Fig. 1B e.g. lane 8 between 32-46 kDa) or APP and PPL (Fig. 1D lane 8 and 10) pull-downs? These species seemingly resemble the peptidyl-tRNA species described below. To exclude that peptidyl-tRNA species are also present in the reactions performed with mRNA containing stop codons I suggest to perform a control, in which Sec61b and PPL translation products derived from stop codon containing mRNA are treated with RNase.

Fig. 2 and RNCs in general

As I understand Materials and Figure Legends, the authors added HisTrx-SGTA to translation reactions primed with non-stop mRNAs and then performed pull-downs employing the His-tag on HisTrx-SGTA without separating RNCs from released chains (see also above) prior to affinity purification. To that end, translation reactions were diluted 5.5-fold in a buffer containing 300 mM NaCl, but no Mg²⁺. Ribosomes are not very stable with low Mg²⁺ and high Na⁺. Moreover, the bound material was eluted with 200 mM imidazole prior to analysis. I am worried that RNCs are not stable under these conditions and nascent chains (even nascent chains still bound to tRNA!) might be released from ribosomes under these conditions. Released chains may bind to HisTrx-SGTA in the course of the experiment.

To make sure that this is not the case I suggest two control experiments:

1. Test if RNCs are stable in buffer A/imidazole.

Generate radiolabelled peptidyl-tRNA/nascent chains, dilute them 5.5 fold with buffer A, incubate for 30 min, add 200 mM imidazole, and then perform ribosome sedimentation via UZ/sucrose cushion. This would more or less mimic the conditions of the pull-down reaction. The control experiment should be performed in parallel with mRNA with and without stop-codon. If the conditions preserve RNCs then the translation product from the stop codon-containing mRNA should be in the supernatant, while peptidyl-tRNAs/nascent chains derived from stop-codon free mRNA should be recovered in the ribosomal pellet.

2. Show that ribosomes/RNCs are bound to HisTrx-SGTA.

The authors should show that the eluted fraction of the IMAC experiments contains a significant amount of ribosomes using antibodies against a ribosomal protein of the 60S as well as one of the 40S subunit.

Fig. 3

Fig. 3B

Of note there is a much larger fraction of peptidyl-tRNA in the TMD-0 lane. Please comment on this observation.

Fig. 3E

Hbs1L depletion works quite well, however, the effect of Hbs1L-depletion is not that convincing. This experiment should be performed in triplicate and should be quantified.

Fig. 4

This is a very important piece of data. In my view it is essential to show that also endogenous SGTA interacts with RNCs carrying nascent ER-membrane proteins!

In contrast to the experiments described above, in which affinity purification was performed via HisTrx-SGTA, in this experiment FLAG-tagged nascent chains were affinity purified via the FLAG-tag. If I understand correctly, the FLAG-pull-down was again performed with the total translation reaction. In this case, one will purify ribosome-bound nascent chains/peptidyl-tRNAs as well as

released chains. It thus remains unclear if SGTA binds post-translationally (to released) or co-translationally (to RNCs). The experiment in Fig. 4 needs to be repeated with RNCs separated from released chains via UZ/sucrose cushion.

I note 35S-labeled FLAG-tagged Sec61b, which is not attached to tRNA has ran out of the gel (compare Fig. 4 to Fig. 3D lane 8 where it is clearly visible).

Fig. 5C

The analysis of the lysines forming crosslinks is not that convincing. The reduction of the crosslink products is only moderate, and in my view, the amount of the totals quite well corresponds with the amount in the IP lanes! This experiment should be performed in triplicate to confirm the conclusion. Alternatively, one could remove this piece of data from the manuscript.

1st Revision - authors' response

3 December 2019

Referee #1:

A small but significant percentage of proteins are delivered to the ER via C-terminal tail-anchor (TA) sequences, and in recent years the machinery required for their targeting and insertion into the ER has been elucidated. In this manuscript, Leznicki and High use a series of biochemical techniques to better characterize one member of the targeting complex, SGTA, which functions with TRC40/Get3, a conserved ATPase that plays a central role in this process. Specifically, Leznicki and High show that SGTA associates with TA proteins as well as with an SRP-dependent membrane protein, a modified form of C99 that harbors an exogenous signal sequence. More surprisingly, SGTA appears to identify proteins with membrane spans prior to the time that a transmembrane domain has emerged into the cytosol. Negative data suggest that Bag6 does not contribute to this association. These data reflect discoveries on the action of SRP in targeting nascent proteins to the ER, i.e., SRP also associates with ribosomes and engages polypeptides prior to the emergence of a transmembrane domain. Moreover, the authors report that SGTA can bind to both a signal sequence and a transmembrane domain, albeit with different efficiencies, in the same protein. Next, Leznicki and High found that the inclusion of microsomes in the *in vitro* translation reaction significantly decreased the extent of SGTA association with the signal sequence-containing C99 protein, and this reaction-performed with this SRP-dependent substrate-also appeared to be independent of Bag6. Finally, consistent with its role as a chaperone-like protein, SGTA protects nascent proteins from the protein quality control machinery, namely ubiquitination, which can otherwise target aberrant nascent proteins for degradation.

Overall, this is an intriguing study that is nicely presented. The quality of the data is high, the results were confirmed using a range of crosslinking/pull-down conditions, and alternative interpretations of the data (e.g., ribosome dissociation) were excluded. Substrates containing specific mutations were also used to refine the conclusions. Nevertheless, it will ultimately be important to complement the results obtained in this study with cell-based systems, such as ribosome profiling combined with SGTA precipitations. This attack is beyond the scope of the current study, but perhaps any caveats from a solely *in vitro* study should be spelled out.

In a related vein, the number of SRP-dependent substrates was limited, and again in the future the use of a higher throughput strategy would better cement the authors conclusions.

*We would like to thank the reviewer for their kind words and appreciation of our work. We agree that a cell-based assays such as ribosome profiling suggested by the reviewer will help obtain more comprehensive insights into the action of SGTA and will aid in identifying its endogenous co-translational substrates. Clearly, this is the next step for our future research. In line with reviewer's suggestion we have now spelled out the limitations of an *in vitro* system and directions for future studies (see page 19, paragraph 2).*

Also, the extent of ubiquitination in Fig. 7B varied significantly, and perhaps the authors can speculate on the reason for this variation and make predictions based on which substrates might exhibit a magnified requirement for SGTA as a nascent chain protector.

We agree with the reviewer that understanding the determinants that define SGTA substrates is the key next step in our work. As suggested by the reviewer (see above) a much broader substrate range

will be needed to identify these common features. However, in order to get some further insight into SGTA-mediated substrate protection we have repeated the quantifications presented in Fig 7B and calculated the intensity of individual ubiquitinated nascent chain species. Together with our new data indicating that SGTA protects a nascent ribosome-bound membrane protein by binding to it rather than actively inducing its deubiquitination (see Figs EV5A and EV5B of the revised manuscript) it seems likely that the extent to which SGTA protects a particular substrate is related to the nascent chain conformation and/or accessibility of any lysine residues within the SGTA-bound nascent chain to an E3 ligase(s). These in turn, could also be affected by the rate of protein synthesis and the kinetics of SGTA binding, and could explain why CD247 modified with three ubiquitin moieties is the only nascent CD247 species significantly affected by the presence of exogenous SGTA (see Fig 7B of the revised manuscript). Our data do suggest that a short distance between two hydrophobic motifs in a nascent chain is not the sole feature that specifies whether a nascent chain is an SGTA substrate and we have now spelled this out in our discussion (see page 19, paragraph 2).

One final open question regards the relationship of other SGTA-associated chaperones and the EMC in these early SRP and SGTA-dependent events. Perhaps this can be discussed or incorporated into the figure.

We thank the reviewer for this suggestion. Based on our pull-down assays using an SGTA variant with mutations in the TPR region (Fig EV1E of the revised manuscript) that disrupts SGTA binding to chaperones of the Hsp70 and Hsp90 families we conclude that these chaperones are not essential for SGTA to recognise its substrates co-translationally. We now spell this out in discussion (see page 17, paragraph 1). Any involvement of the EMC in SGTA binding to the RNCs is more speculative and we have now suggested that the EMC, along the SRP receptor and the Sec61 complex, is a potential candidate for a membrane-localised component that mediates membrane-dependent release of SGTA (see page 18, paragraph 4 and page 19, paragraph 1). Further experiments will be needed to establish if this release is indeed protein-mediated and to investigate any potential candidates.

Other comments:

1. The authors should include a Table or text to define the substrates used in this paper. What is RELT, F11R, etc?

The abbreviations of the TA-proteins used in our study have now been explained in the legend to Fig 1. C99 variants have already been described in the manuscript text whilst substrates used in the original Fig EV4 (Fig EV3 in the revised version of the manuscript) are now defined in the figure legend.

2. In Fig. EV2, please define the arrowhead and arrow.

We thank the reviewer for pointing out our omission. Fig EV2 has now been moved and is presented as Appendix Fig S1B, with both the arrow and arrowhead defined.

Referee #2:

Comments to the Author

EMBOR-2019-48835V1
Leznicki and High,

"SGTA associates with nascent membrane protein precursors."

How newly made transmembrane proteins are accurately assembled into the ER membrane? This point has been receiving the most attention as an important issue for a long time of decades. In this manuscript, Drs. Leznicki and High provided a new insight on this issue from the viewpoint of SGTA, a component of cytosolic protein quality control machinery.

Using an in vitro translation system, the authors investigated the interactions of SGTA with newly made membrane protein precursors. They showed that SGTA is capable of recognizing a range of nascent membrane proteins. Amongst the translation products recognized by SGTA were peptidyl-tRNA species that represent ribosome-associated nascent chains. Notably, SGTA recruitment to the

ribosome relies on the existence of TMD even when the hydrophobic sequence is still located within the ribosomal exit tunnel, possibly through inducing structural rearrangements of ribosomes. SGTA binds directly to the exposed hydrophobic signal and prevent the cytosolic exposure of TMDs, and thereby prevent the ubiquitination and/or aggregation of these nascent chains until they engage the ER translocon. Indeed, SGTA dissociate from the nascent chain when the client was successfully inserted into the ER lipid bilayer. Such a previously unanticipated recruitment of SGTA to the ribosome would ensure that it has early access to client membrane proteins which may in turn help to increase the fidelity of their post-translational delivery to the ER.

I found this manuscript contains really interesting ideas that how SRP and SGTA collaborate together to support the successful synthesis of multiple membrane spanning proteins. With full of insightful viewpoint, I found the publication of this manuscript at this moment is really timely. Therefore, I recommend this manuscript for publication after adding some experiments to support their main claims. My specific comments were provided as follows.

We thank the reviewer for finding our work interesting and recommending its publication.

Specific comments;

(1)

The authors proposed that SGTA recognizes ribosome-associated transmembrane protein precursors to prevent the ubiquitination and/or aggregation of these nascent chains until they engage to the ER. I found this idea very interesting. To further support the proposed function of SGTA, I recommend to immunodeplete the endogenous SGTA protein from RRL with their anti-SGTA antibody. Since RRL contains endogenous SGTA (as shown in Figs 6B and EV2), and authors had performed immunodepletion of Hbs1L (Fig. 3E) and BAG6 (Fig. EV5A) from RLL, these points can be clarified relatively easily. As the idea is so important, I believe that SGTA depletion should be necessary before claiming SGTA is critical for quality control of nascent membrane proteins.

We agree with the reviewer that depletion experiments would add additional weight to our findings. However, while revising our manuscript, and in order to address points raised by reviewer 3, we have discovered that RRL contains two SGT proteins, SGTA (not annotated in the current rabbit genome) and SGTB (annotated as G1SX57). Based on our new mass spectrometry data (see Appendix Figs S2B-S2D), rabbit SGTA is the main interacting partner of the substrates studied. However, SGTB also binds to immobilised Sec61b (Appendix Fig S2 of the revised manuscript), and likely other hydrophobic substrates, indicating some functional redundancy between SGTA and SGTB. Such redundancy is further supported by additional new experiments (Figs EV5C and EV5D of the revised manuscript) which show that recombinant rabbit and human SGTBs can also inhibit ubiquitination of a model nascent ribosome-bound membrane protein. These new findings therefore complicate the interpretation of any results that are based solely on SGTA immunodepletion. We believe that the simultaneous immunodepletion of both rabbit SGTA and SGTB will be a more informative approach and we will strive to achieve this goal. Unfortunately, at the present time we lack the tools necessary to immunodeplete rabbit SGTB. At the same time, the fact that endogenous SGTA co-purifies with RNCs translating Sec61b (Fig 4) and can be cross-linked to stalled PPL-C99^{FL} (Figs 6B, EV2B and Appendix Fig S1B of the revised manuscript) all suggest that endogenous rabbit SGTA behaves in the same way as the recombinant human protein, i.e. they perform the same function.

(2)

In Fig 7A, the description in the text does not correspond well with the actual data. The authors noted that the inclusion of exogenous SGTA significantly reduced the amount of ubiquitinated precursor (p14, line 27). However, I can see a very small decrease of ubiquitinated substrates with SGTA addition, both in the case in F11R and TMEM174. Are these small differences of polyubiquitination of the membrane protein physiologically relevant? Why the co-translational ubiquitination of CD247 was not affected significantly by the addition of HT-SGTA, even though two hydrophobic signals (SS and TMD) of this protein are separated by a short linker as in the case of TMEM174?

In order to address reviewer's comment, we have re-quantified data presented in Fig 7A and calculated the change in intensity of individual ubiquitinated nascent chain species upon SGTA addition. Again, our analysis indicates that changes in the ubiquitination of F11R and TMEM174 are statistically highly significant. We believe that these changes are physiologically relevant as, for example for TMEM174, we observe ~70% reduction in nascent chain ubiquitination.

We thank the reviewer for pointing out that the distance between two hydrophobic motifs within a nascent chain is not the only determinant that defines SGTA substrates. Our new data indicate that SGTA protects a nascent ribosome-bound membrane protein by binding to it rather than actively inducing its deubiquitination (see Figs EV5A and EV5B of the revised manuscript) and it therefore seems likely that the extent to which SGTA protects its substrates is related to the nascent chain conformation and accessibility of lysine residues of SGTA-bound nascent chain to an E3 ligase(s). These factors could also be related to nascent chain translation rate and the kinetics of SGTA binding, and could explain why CD247 modified with three ubiquitin moieties is the only nascent CD247 species significantly affected by the presence of exogenous SGTA (see Fig 7B of the revised manuscript). We agree that a short distance between two hydrophobic motifs in a nascent chain does not seem to be the sole feature that specifies whether a nascent chain is an SGTA substrate and we have now spelled this out in our discussion (see page 19, paragraph 2).

(3)

Related to my previous comment, authors quantified the accumulations of ubiquitinated precursor species derived from several membrane proteins in Fig 7A and 7B. I would like to ask you whether this assay includes inhibitors for proteasomes and deubiquitination enzymes in the RLL. Since RLL contains large amounts of these enzymes, the presence or the absence of these inhibitors might affect the interpretation of the results greatly. I recommend to add these inhibitors in the RLL and precipitation buffer for these kind of polyubiquitin pull-down assay.

We thank the reviewer for their valuable comment. The RLL that we used had been optimised for in vitro protein translation, which involves adding hemin to allow for efficient translation initiation. Hemin is also an inhibitor of the proteasome (Hoffman, L. and Rechsteiner, M. (1996). *J. Biol. Chem.* 271, 32538-32545; Oberdorf, J., Carlson, E. and Skach, W. (2001). *Biochemistry* 40, 13397-13305) which explains why translation-grade RLLs are inefficient in mediating protein degradation. Hence, the contribution of proteasome-mediated degradation is negligible. As the reviewer pointed out RLL has high ubiquitination and deubiquitination activities. For this reason, all samples that relate to substrate ubiquitination were treated with NEM at the end of the translation reaction. NEM modifies cysteine residues and hence would irreversibly inactivate E1 and E2 enzymes (hence blocking any on-going ubiquitination) and all cysteine-based deubiquitinating enzymes (DUBs) which constitute a vast majority of all DUBs. Hence, we believe that the data we present accurately reflect steady-state ubiquitination status of the substrates used. Furthermore, based on the reviewer's comment we have also repeated the ubiquitination reactions in the presence of ubiquitin aldehyde (Ub-Ald), a specific inhibitor of cysteine-based DUBs. These results are presented in Figures EV5A and EV5B of the revised manuscript and allow us to conclude that SGTA most likely acts by sterically blocking access of an E3 ligase(s) to exposed TMDs (see page 16, paragraph 2). We thank Referee 2 for this suggestion.

Other minor points:

(4)

It is helpful if authors indicate the position of 11 kDa marker in the Fig. 1, as like in Figs 3E and 6, since RAMP4 and Sec6a beta migrated around this size.

We checked the original data for this experiment but unfortunately the 11 kDa marker on the two gels showing the TA-proteins in Fig 1 was not visible on the autoradiographs. This could be either because it had run just off the gel or it had not been marked with the diluted solution of the isotope that we use to visualise pre-stained protein markers on autoradiographs.

(5)

What is "x" in xSRP54 in Fig 5 and Fig 6? I assume these mean "cross linked with PPL-C99", since the sum of the molecular weights of SRP54 and PPL-C99 gives that of xSRP54. Please define the exact meaning of this abbreviation in the corresponding figure legends.

We thank the reviewer for pointing that we had not defined this symbol. We have defined "x SRP54", "x HisTrx-SGTA" and "x endo. SGTA" in legends to Figures 5, 6 and EV2 of the revised manuscript.

(6)

The data in Fig. 5C need quantifications. Can you see statistically significant differences between K4/9R and K48-87R?

We thank the reviewer for this comment. We have now repeated this experiment and carried out statistical analysis which indeed confirmed our earlier qualitative conclusions that lysines surrounding substrate TMD form the principal (albeit not exclusive) cross-linking partners with SGTA.

(7)

In the reference list, I found a number of style inconsistency (all characters were capitalized in Ref. 40, for an example). Please amend them.

We thank the reviewer for their careful analysis of the reference list. We have used EndNote X8 with EMBO Reports referencing style to ensure that any mistakes have now been corrected. Regarding Ref. 40 all characters are capitalised because that is how the title appeared in the original publication in 1964.

Referee #3:

Leznicki and High report interesting findings, which shed new light on the ER targeting function of SGTA. Most importantly, the authors present data, which show that SGTA i) can interact with ribosome-bound nascent chains (RNCs) of ER-targeted proteins containing C-terminal tail anchors or internal transmembrane domains (TMDs); ii) binds to these RNCs even when the TMD is still inside the ribosomal tunnel; and iii) may interact with RNCs, even if SRP is bound.

These findings are novel and have important implications for the whole ER targeting field.

However, there are two general problems with the experimental set up, which, in my view, need to be clarified to justify the conclusions drawn by the authors.

1. It is not firmly established that SGTA binds to ribosome-bound nascent chains, rather than to released nascent chains. Studies analysing the binding of factors to RNCs standardly sediment RNCs to remove any spontaneously released polypeptides prior to further analysis. Such unwanted release occurs inevitably in every translation system! This sedimentation step is missing in the experiments performed in this study. This is a critical issue that needs to be addressed.

We thank the reviewer for this comment. In order to address this point and subsequent related points raised by the reviewer (see below) on we have carried out a number of additional experiments to ensure that we are looking at intact RNCs. These are:

- a) We have combined HisPur Cobalt pull-downs with a sedimentation step, which clearly shows that material bound to HisTrx-SGTA and eluted at high imidazole concentration can be pelleted through a sucrose cushion indicative of it being part of a large complex such as an RNC. We have carried out this analysis both for the C99 variants and TA-proteins used in this study. This new data is presented in Fig EVIC of the revised manuscript.
- b) We have repeated the pull-down assay using rabbit reticulocyte lysate (RRL) depleted of the Hbs1L splitting factor a total of 4 times. We then quantified the tRNA-bound species of stalled FLAG-Sec61b in the input and HisTrx-SGTA pull-down fractions relative to control-depleted RRL (Fig 3E of the revised manuscript). We can see an almost two-fold increase in the amount of tRNA-bound FLAG-Sec61b species recovered with HisTrx-SGTA from Hbs1L-depleted RRL and this difference is statistically significant. If SGTA bound only released nascent chains or nascent chains on partially disassembled ribosomes rather than intact RNCs then one would expect to see a reduction in the amount of co-isolated tRNA-bound nascent chains upon Hbs1L depletion because the latter manipulation prevents ribosome splitting and hence stabilises RNCs. Such RNC stabilisation was in turn confirmed by quantifying tRNA-bound species in total translation reactions – “input” panels in Figs 3D and 3E of the revised manuscript.
- c) We have repeated the experiment presented in Fig 4 and added an additional sedimentation step through a sucrose cushion as suggested by the reviewer (see below). In short, FLAG-tagged Sec61b WT or its 3R variant were translated in RRL from an mRNA lacking a stop codon, Sec61b variants isolated on anti-FLAG beads, bound proteins eluted with the 3xFLAG peptide and the eluate centrifuged through a sucrose cushion. This last step, as suggested by Referee 3, ensures that we are looking at intact RNCs. Importantly, we further confirmed the recovery of intact RNCs by Western blotting with antibodies against proteins

of both the 60S (anti-Rpl17) and 40S (anti-Rps3) ribosomal subunits. Crucially, we detected endogenous SGTA in the pelleted fractions from WT Sec61b sample but not the 3R variant, confirming its association with intact 80S ribosomes translating hydrophobic nascent chains.

- d) We apologise for not making this clear in our original manuscript but following cross-linking samples were routinely subjected to ultracentrifugation through a sucrose cushion in order to recover intact RNCs and only these pelleted fractions were later used either for direct analysis or immunoprecipitation with anti-SRP54 or anti-SGTA antibodies (Figs 5, 6, EV2, EV3H, EV3I, EV4 and Appendix Fig S1B of the revised manuscript). Similarly, the functional assay addressing the role of SGTA on co-translational ubiquitination involves an ultracentrifugation step through a sucrose cushion, again to make sure that we analysed co-translational ubiquitination that was occurring on RNCs (Figs 7 and EV5 of the revised manuscript).

2. There needs to be some evidence that tagged human SGTA (HisTrx-SGTA) is able to functionally replace untagged rabbit SGTA. The authors state that the two proteins are not highly homologous. Possibly HisTrx-SGTA, which is added to translation reactions in excess when compared to the other GET pathway components, does "something" endogenous SGTA, which is bound to partner proteins/chaperones, would never do? In this context, it is essential to improve Fig. 4 (see also below) and Fig. EV2, which should become part of the main part of the manuscript. If I understand correctly the authors suggest that chicken anti-SGTA can IP rabbit SGTA? This should be confirmed via a control blot. Should be possible using the anti-human SGTA AB, which seemingly recognizes rabbit SGTA on Western Blots (Fig. 4). A control blot or appropriate reference should be added to show that the relevant band in Fig. 4 is indeed rabbit SGTA.

We thank the reviewer for bringing this issue to our attention. We have now improved Fig 4 (see above) based on the reviewer's comments and Fig EV2 of the original manuscript (Appendix Fig S1B of the revised manuscript). We also carried out a validation of the chicken anti-SGTA antibody to show that the endogenous protein that co-purifies with FLAG-Sec61b RNCs in Fig 4 and that can be cross-linked to stalled PPL-C99 (Figs 6B, EV2B and Appendix Fig S1B of the revised manuscript) is indeed SGTA. This was done by Western blotting with chicken anti-SGTA antibody pre-incubated with an excess of purified recombinant HisTrx-SGTA against human HeLa cell lysate and RRL (Appendix Fig S2A of the revised manuscript). The results clearly indicate that in both HeLa cell lysate and RRL there is a single species recognised by chicken anti-SGTA antibody that no longer reacts with anti-SGTA antibody once it has been blocked with recombinant HisTrx-SGTA.

Referee 3's comments prompted us to revisit the surprising absence of a bona fide SGTA in rabbit, especially since the closest annotated orthologue, G1SX57, is actually almost identical to human SGTB (see Appendix Fig S3B). We therefore carried out mass spectrometry based identification of the endogenous rabbit protein specifically recognised by the chicken anti-SGTA antibody that we used to show association of endogenous rabbit SGT with nascent chains (Figs 4, 6B and EV2B, lane 7). This analysis showed that this ~38 kDa protein contains peptides that can be unambiguously assigned to human SGTA but not rabbit G1SX57 (Appendix Figs S2B and S2C). As an alternative approach we immobilised bacterially expressed and purified Sec61b full-length and its variant lacking the TMD on solid support, and used these immobilised proteins to carry out pull-downs from RRL (Appendix Fig S2D). We had previously used this method to successfully identify Bag6 as a biogenesis factor for TA-proteins (Leznicki et al., (2010) Journal of cell science 123: 2170-8). We then carried out mass spectrometry based identification of the ~38 kDa protein specifically eluted with 0.1% (w/v) Triton X-100. Again, when the peptides were run against a human protein database the major protein identified was human SGTA, and once again the peptides were unique to SGTA and distinct from the sequence of G1SX57 (Appendix Fig S2B). Interestingly, in the latter experiment the same sample also contained peptides that correspond directly to G1SX57 (Appendix Figs S2B and S2D). On the basis of these findings we now present multiple sequence alignment of SGTA from various mammalian species, which shows that the protein is highly conserved even in evolutionarily distant mammalian species such as human, mouse, whales, bats and Tasmanian devil. Importantly, these species all have clear orthologues of both SGTA and SGTB. This includes *Ochotona princeps* (American pika), which like rabbit belongs to taxonomic order Lagomorpha. Furthermore, Itakura et al. (2016, Molecular Cell 63: 21-33) also identified SGTA and not G1SX57 as an interacting partner of a mitochondrial membrane protein synthesised in RRL. Taking all of these data together, we now propose that the simplest and most likely explanation for our findings is

that rabbits have SGTA and SGTB but that to date only the SGTB orthologue has been annotated at the genome level. Clearly, this issue requires further clarification but this will require analysis by researchers with expertise distinct from our own and is beyond the scope of our current study. Nevertheless, the presence of a close rabbit orthologue of human SGTA supports our decision to use recombinant human SGTA during the course of our studies.

Furthermore, we now show that like human SGTA, both G1SX57 (rabbit SGTB) and human SGTB can also limit the co-translational ubiquitination of a nascent membrane protein (Figs EV5C and EV5D of the revised manuscript). This potential functional redundancy between SGTA and SGTB is something that we plan to explore further in the future.

Some more comments and suggestions are detailed below.

Fig. 1

The data convincingly show the well characterized post-translational interaction between SGTA and ER-targeted proteins. With respect to the data shown below, however, I wonder what is the larger species detected in the Synb2, Sec61b (Fig. 1B e.g. lane 8 between 32-46 kDa) or APP and PPL (Fig. 1D lane 8 and 10) pulldowns? These species seemingly resemble the peptidyl-tRNA species described below. To exclude that peptidyl-tRNA species are also present in the reactions performed with mRNA containing stop codons I suggest to perform a control, in which Sec61b and PPL translation products derived from stop codon containing mRNA are treated with RNase.

We have carried out the suggested experiment with RNaseA treatment. However, we failed to reproducibly detect these species in the HisTrx-SGTA pull-down fractions. Based on the electrophoretic mobility of these species it is possible that they correspond to a small fraction of nascent chains whose translation termination might have been incomplete. In agreement with this interpretation, translation termination pausing has been observed for both Sec61b and Syb2 (also known as VAMP2) but not for most other TA-proteins (Mariappan et al. (2010) Nature 466: 1120-4; Ingolia et al. (2011) Cell 147: 789-802).

Fig. 2 and RNCs in general

As I understand Materials and Figure Legends, the authors added HisTrx-SGTA to translation reactions primed with non-stop mRNAs and then performed pull-downs employing the His-tag on HisTrx-SGTA without separating RNCs from released chains (see also above) prior to affinity purification. To that end, translation reactions were diluted 5.5-fold in a buffer containing 300 mM NaCl, but no Mg²⁺. Ribosomes are not very stable with low Mg²⁺ and high Na⁺. Moreover, the bound material was eluted with 200 mM imidazole prior to analysis. I am worried that RNCs are not stable under these conditions and nascent chains (even nascent chains still bound to tRNA!) might be released from ribosomes under these conditions. Released chains may bind to HisTrx-SGTA in the course of the experiment.

To make sure that this is not the case I suggest two control experiments:

1. Test if RNCs are stable in buffer A/imidazole.

Generate radiolabelled peptidyl-tRNA/nascent chains, dilute them 5.5 fold with buffer A, incubate for 30 min, add 200 mM imidazole, and then perform ribosome sedimentation via UZ/sucrose cushion. This would more or less mimic the conditions of the pull-down reaction. The control experiment should be performed in parallel with mRNA with and without stop-codon. If the conditions preserve RNCs then the translation product from the stop codon-containing mRNA should be in the supernatant, while peptidyl-tRNAs/nascent chains derived from stop-codon free mRNA should be recovered in the ribosomal pellet.

We are very grateful to the reviewer for suggesting this experiment. When we have carried out this experiment as described we did indeed find that the dilution of translation reactions in a buffer containing high NaCl and no additional Mg²⁺ substantially disrupted the integrity of the RNCs when compared to dilution in a low salt buffer that contains Mg²⁺. This control experiment also showed that the inclusion of 5 mM MgCl₂ in our original high salt buffer preserves the integrity of RNCs to levels qualitatively identical to dilution in the low salt buffer that contains Mg²⁺. In light of this new information we have now REPEATED ALL PULL-DOWN EXPERIMENTS where mRNAs lacking a stop codon were used. These include data presented in Figs 2, 3B, 3C, EV1, EV3B-G of the revised manuscript. The data included in Fig 3E of the original manuscript (Figs 3D and 3E of the revised manuscript) were not repeated since they had already been carried out using buffer

supplemented with 5 mM MgCl₂. All of our new data are qualitatively the same as our previous findings. Moreover, as a proof of RNCs integrity we now include Fig EV1C where a HisPur Cobalt pull-down was combined with sedimentation analysis, corresponding to the experiment suggested by Referee 3. Additional data which further support our proposal that we are studying intact RNCs are also detailed above, and we thank Referee 3 for bringing this point to our attention and helping us to improve the quality of our data accordingly.

2. Show that ribosomes/RNCs are bound to HisTrx-SGTA.

The authors should show that the eluted fraction of the IMAC experiments contains a significant amount of ribosomes using antibodies against a ribosomal protein of the 60S as well as one of the 40S subunit.

Based on the reviewer's comment to Fig 4 we have repeated the experiment presented there and included an additional step of ultracentrifugation through a sucrose cushion. We could detect both the 40S and 60S ribosomal subunits in the pellet fraction that also contained nascent chains and SGTA that selectively co-sedimented with the RNCs containing a membrane protein. This experiment provides an elegant proof of RNC integrity using a complementary approach to that detailed in our response to point 1 above.

Fig. 3

Fig. 3B

Of note there is a much larger fraction of peptidyl-tRNA in the TMD-0 lane. Please comment on this observation.

We have repeated this experiment using the new pull-down conditions suggested by Referee 3 (see above) and found that the previous difference in peptidyl-tRNA species observed with the PPL^{ssKO}-C99 TMD+0 sample is no longer evident and comparable levels of peptidyl-tRNA species are observed for other PPL^{ssKO}-C99 variants such as TMD+20 (see Figs 3B and EV1A, cf. lanes 2 and 4).

Fig. 3E

Hbs1L depletion works quite well, however, the effect of Hbs1L-depletion is not that convincing. This experiment should be performed in triplicate and should be quantified.

We thank the reviewer for this suggestion. We have repeated this experiment four times and carried out statistical analysis. These new results clearly indicate that there is a statistically significant effect of Hbs1L-depletion on the total amount of tRNA-bound species which confirms that depletion of Hbs1L manifests itself at the functional level by inhibiting ribosome splitting. Quantification of the SGTA associated tRNA-bound species also shows a statistically significant increase upon Hbs1L depletion. This in turn strongly supports our data that SGTA binds to intact 80S ribosomes as otherwise one would expect to see reduced binding of SGTA to tRNA-bound species that accumulate as a result of Hbs1L depletion.

Fig. 4

This is a very important piece of data. In my view it is essential to show that also endogenous SGTA interacts with RNCs carrying nascent ER-membrane proteins!

In contrast to the experiments described above, in which affinity purification was performed via HisTrx-SGTA, in this experiment FLAG-tagged nascent chains were affinity purified via the FLAG-tag. If I understand correctly, the FLAG-pull-down was again performed with the total translation reaction. In this case, one will purify ribosome-bound nascent chains/peptidyl-tRNAs as well as released chains. It thus remains unclear if SGTA binds post-translationally (to released) or co-translationally (to RNCs). The experiment in Fig. 4 needs to be repeated with RNCs separated from released chains via UZ/sucrose cushion.

We are grateful to the reviewer for this comment. Based on the reviewer's suggestion we have now repeated the experiment presented in Fig 4 and added an ultracentrifugation step where material eluted from anti-FLAG beads with the 3xFLAG peptide was spun down through a sucrose cushion. Our results show that SGTA co-sediments with intact ribosomes translating wild-type Sec61b but not its 3R variant even though the recovery of the nascent chains and ribosomes (identified by immunoblotting with antibodies against proteins of the 60S and 40S ribosomal subunits) is comparable. Additional support for the interaction of ribosome-stalled nascent chains with

endogenous SGTA is presented in Figs 6B and EV2B, lane 7, as well as in Appendix Fig S1B of the revised manuscript which show cross-linking adduct formation between stalled PPL-C99 and endogenous SGTA.

I note 35S-labeled FLAG-tagged Sec61b, which is not attached to tRNA has ran out of the gel (compare Fig. 4 to Fig. 3D lane 8 where it is clearly visible).

In the new version of the figure both tRNA-bound and unbound Sec61b species are clearly visible.

Fig. 5C

The analysis of the lysines forming crosslinks is not that convincing. The reduction of the crosslink products is only moderate, and in my view, the amount of the totals quite well corresponds with the amount in the IP lanes! This experiment should be performed in triplicate to confirm the conclusion. Alternatively, one could remove this piece of data from the manuscript.

We thank the reviewer for this comment. We have now repeated this experiment in triplicate and carried out statistical analysis where the relative intensity of the cross-linking adducts were normalised to total translation products from the corresponding input lanes and subsequently compared to the values obtained for the wild-type protein. These data are presented in Fig 5D of the revised manuscript and confirm our earlier qualitative conclusions that lysines surrounding substrate TMD are the primarily (albeit not exclusive) cross-linking partners with SGTA.

2nd Editorial Decision

24 February 2020

Please accept my apologies for this very unusual delay. We have finally received the second report on your manuscript. Since this referee had raised many critical points, I had decided to wait for it. As you will see from the reports below, both referees are now all positive about its publication in EMBO reports. I am therefore writing with an 'accept in principle' decision, which means that I will be happy to accept your manuscript for publication once a few minor issues/corrections have been addressed.

REFEREE REPORTS

Referee #2:

In this revision, authors have addressed most of the concerns appropriately. Their additional experiments revealed that there is a functional redundancy between SGTA and SGTB in RLL, and this makes immuno-depletion experiments difficult at this moment. Authors have added new data and corresponding discussion to improve the manuscript greatly. Their additional experiments support the validity of their findings. I believe that the manuscript is now suitable for publication.

Referee #3:

I carefully looked at the response to my concerns and I thank the authors for taking my comments and suggestions seriously. Also, I have looked at the revised version of the manuscript, with a focus on the changes connected to my comments. I find the experimental data significantly improved. Especially the new version of Figure 4 is convincing. The only thing ... I do not quite understand how the outcome of the pull down experiments of RNCs with Mg²⁺ can be "qualitatively the same" as without Mg²⁺, when RNCs fell apart. But anyway: I really find the work convincing and the finding that SGTA can interact with nascent membrane protein precursors novel and very interesting. This result will significantly impact on the protein targeting field.

In my opinion the work should be published in EMBO Reports.

2nd Revision - authors' response

25 February 2020

The authors performed all minor editorial changes.

Corresponding Author Name: Stephen High

Manuscript Number: EMBOR-2019-48835